# Improved Theoretically-Grounded Evolutionary Algorithms for Subset Selection with a Linear Cost Constraint

**Dan-Xuan Liu** [1] [2]  **Chao Qian** [1] [2]

## Abstract

The subset selection problem with a monotone and submodular objective function under a linear cost constraint has wide applications, such as maximum coverage, influence maximization, and feature selection, just to name a few. Various greedy algorithms have been proposed with good performance both theoretically and empirically. Recently, evolutionary algorithms (EAs), inspired by Darwin's evolution theory, have emerged as a prominent methodology, offering both empirical advantages and theoretical guarantees. Among these, the multi-objective EA, POMC, has demonstrated the best empirical performance to date, achieving an approximation guarantee of $(1/2)(1 - 1/e)$. However, there remains a gap in the approximation bounds of EAs compared to greedy algorithms, and their full theoretical potential is yet to be realized. In this paper, we re-analyze the approximation performance of POMC theoretically, and derive an improved guarantee of $1/2$, which thus provides theoretical justification for its encouraging empirical performance. Furthermore, we propose a novel multi-objective EA, EPOL, which not only achieves the best-known practical approximation guarantee of 0.6174, but also delivers superior empirical performance in applications of maximum coverage and influence maximization. We hope this work can help better solving the subset selection problem, but also enhance our theoretical understanding of EAs.

## 1. Introduction

The subset selection problem aims to select a subset $X$ from a given ground set $V$ that maximizes a specific func-

tion $f$, while the cost of the subset $X$ must not exceed a given budget $B$. This fundamental problem is NP-hard and arises in diverse domains involving cost-aware resource allocation, including combinatorial optimization, computer networks, data mining, and machine learning. Potential applications include maximum coverage (Feige, 1998), maximizing coverage under budget constraints; influence maximization (Kempe et al., 2003), maximizing social influence spread under budget constraints; sensor placement, balancing information gain with installation costs (Krause et al., 2006); recommendation systems, promoting products within advertising budgets while respecting user preferences (Ashkan et al., 2015); unsupervised feature selection, optimizing reconstruction error of a data matrix under feature resource constraints (Feng et al., 2019); active learning, selecting maximally informative data samples under limited annotation budgets (Golovin & Krause, 2011); and human-assisted learning, optimizing machine learning models with limited expert resources (De et al., 2020; Liu et al., 2023).

By introducing monotonicity and submodularity, the algorithms with theoretical guarantees for subset selection have been well studied. A set function $f$ is monotone (typically non-decreasing) if it does not decrease with the addition of items to a set. A set function $f$ is submodular if $\forall X \subseteq Y, v \notin Y : f(X \cup v) - f(X) \geq f(Y \cup v) - f(Y)$, implying the diminishing returns property (Nemhauser et al., 1978). In this paper, we focus on the subset selection problem as follows:

$$\arg\max_{X \subseteq V} f(X) \quad \text{s.t.} \quad c(X) = \sum_{v \in X} c(v) \leq B, \quad (1)$$

where the objective function $f : 2^V \to \mathbb{R}$ is monotone and submodular, and the cost function $c : 2^V \to \mathbb{R}$ is linear. For the special case $c(X) = |X| \leq B$, a simple greedy algorithm achieves the optimal polynomial-time approximation guarantee of $1 - 1/e$ (Nemhauser et al., 1978).

For the general problem presented in Eq. (1) with a linear cost constraint, greedy algorithms are also mainstream algorithms, and many variants have been proposed. The Generalized Greedy Algorithm (GGA) iteratively selects an item maximizing the ratio of the increment on $f$ and $c$, achieving a $(1/2)(1 - 1/e) \approx 0.32$-approximation ratio (Krause & Guestrin, 2005), which was further improved

[1]National Key Laboratory for Novel Software Technology, Nanjing University, Nanjing 210023, China [2]School of Artificial Intelligence, Nanjing University, Nanjing 210023, China. Correspondence to: Chao Qian <qianc@nju.edu.cn>.

*Proceedings of the 42^{nd} International Conference on Machine Learning*, Vancouver, Canada. PMLR 267, 2025. Copyright 2025 by the author(s).

to $1 - 1/\sqrt{e} \approx 0.39$ (Lin & Bilmes, 2010). Another greedy algorithm, Greedy$^+$, extends GGA by considering additional single-item expansions of all intermediate solutions, achieving a better approximation ratio of 0.5 (Yaroslavtsev et al., 2020). The 1-guess-Greedy$^+$ variant further improves this approach, reaching an impressive approximation ratio of 0.6174, by merely executing a single partial enumeration step. While there are greedy algorithms that obtain the theoretical optimal approximation of $1-1/e$, they are generally impractical due to high computational costs. Sviridenko (2004) developed a $(1 - 1/e)$-approximation algorithm by selecting three optimal elements and using a greedy approach, which, however, has a high time complexity of $O(n^5)$, where $n = |V|$ is the size of the ground set $V$. Later, Badanidiyuru & Vondrák (2014) and Ene & Nguyen (2019) proposed greedy-based algorithms that achieve a $(1 - 1/e - \epsilon)$-approximation, where $\epsilon > 0$. However, their computational costs remain prohibitively high, with time complexities of $O(n^2(\frac{\log n}{\epsilon})^{O(1/\epsilon^8)})$ and $(1/\epsilon)^{O(1/\epsilon^4)} n \log^2 n$, respectively. These complexities render the algorithms impractical for real-world applications. For additional unique algorithmic approaches, please refer to (Tang et al., 2022; Kulik et al., 2021; Li et al., 2022).

Since greedy algorithms may get trapped in local optima, evolutionary algorithms (EAs), inspired by Darwin's theory of evolution, have been recently applied to subset selection (Zhou et al., 2019). They are general-purpose randomized heuristic optimization algorithms (Bäck, 1996) that mimic variational reproduction and natural selection. Starting from an initial population of solutions, EAs iteratively improve solutions by employing global-search operators such as mutation. They have achieved superior empirical performance than greedy algorithms, while also ensuring theoretical guarantees. Qian et al. (2017a) proposed a simple multi-objective EA named POMC to maximize $f$ and minimize $c$ simultaneously, which can achieve an approximation ratio of $(1/2)(1 - 1/e)$, using $O(nBP_{max}/\delta)$ expected running time[1], where $P_{max}$ is the largest size of population during the running of POMC, and $\delta = \min_{v \in V} c(v)$ is the minimum cost of any item in $V$. EAMC (Bian et al., 2020) and EVO-SMC (Zhu et al., 2024) are single-objective EAs for subset selection. They are guided by surrogate functions of integrating $f$ and $c$, and maintain a limited number of solutions for each possible subset size. They ensure approximation ratios of $(1/2)(1 - 1/e)$ and 0.5, respectively, using $O(n^2 K_B)$ expected running time, where $K_B$ denotes the largest size of a subset satisfying the constraint $c(X) \le B$. FPOMC (Bian et al., 2021) modifies POMC by introducing a greedy selection strategy and also achieves a $(1/2)(1 - 1/e)$-approximation ratio with an expected run-

[1]The expected running time refers to the expected number of evaluations of the objective function, as objective evaluation is usually the most expensive part of the algorithmic process.

ning time of $O(n^2 K_B)$. For more EAs with theoretical guarantees for a cost function mapping to non-negative integers instead of reals, i.e., $c : 2^V \to \mathbb{N}$, please see (Neumann & Witt, 2023; Neumann & Rudolph, 2024). Among these EAs, POMC achieves the best experimental results (Roostapour et al., 2022). Table 1 provides a summary of the practical algorithms for the subset selection problem considered in Eq. (1). Note that there have also developed a series of EAs with theoretical guarantees for diverse variants of subset selection, e.g., (Qian et al., 2015a; 2016; 2017b;c; 2018; 2020; Bian et al., 2022; Liu & Qian, 2024; Qian et al., 2017d; 2019; Qian, 2020; 2021; Qian et al., 2022; 2023)

Despite exhibiting the best empirical performance, POMC still has gaps in approximation ratios compared to other algorithms. Thus, an interesting question is whether the currently-known approximation ratio of POMC is tight. To further explore the potential of EAs, can we design EAs better than POMC both theoretically and empirically? In this paper, we try to address these questions and make the following contributions:

- Through a refined analysis of the lower bound for improving $f(X)$ by adding a specific item to $X$, we improve the approximation ratio of POMC from $(1/2)(1 - 1/e)$ to $1/2$ in Theorem 3.1.

- We propose a new multi-objective EA called EPOL, which achieves an approximation ratio of 0.6174 (Theorem 4.1), using $O(n^2 BP_{max}/\delta)$ expected running time. EPOL is easily parallelizable, and the expected running time can be reduced to $O(nBP_{max}/\delta)$, the same as that of POMC.

- Experiments across various settings of the applications of maximum coverage and influence maximization validate the best performance of POMC among all existing algorithms, and show that EPOL can further improve the performance significantly in almost all cases.

## 2. Preliminaries

Let $\mathbb{R}$ and $\mathbb{R}^+$ denote the set of reals and non-negative reals, respectively. The set $V = \{v_1, v_2, \dots, v_n\}$ denotes a ground set. A set function $f : 2^V \to \mathbb{R}$ is monotone if $\forall X \subseteq Y : f(X) \le f(Y)$. The submodularity represents the diminishing returns property (Nemhauser et al., 1978), i.e., adding an item to a set $X$ gives a larger benefit than adding the same item to a superset $Y$ of $X$. A set function $f : 2^V \to \mathbb{R}$ is submodular if $\forall X \subseteq Y \subseteq V, v \notin Y$,

$$f(X \cup v) - f(X) \ge f(Y \cup v) - f(Y); \quad (2)$$

or equivalently for any $X \subseteq Y \subseteq V$,

$$f(Y) - f(X) \le \sum_{v \in Y \setminus X} \big( f(X \cup v) - f(X) \big). \quad (3)$$

*Table 1.* Summary of practical algorithms (with approximation guarantees and running time) for the subset selection problem in Eq. (1).

| | Algorithm | Guarantee | Running time |
|---|---|---|---|
| **Greedy algorithms** | GGA (Krause & Guestrin, 2005) | $(1/2)(1 - 1/e) \approx 0.32$ | $O(nK_B)$ |
| | GGA (Lin & Bilmes, 2010) | $1 - 1/\sqrt{e} \approx 0.39$ | $O(nK_B)$ |
| | Greedy$^+$ (Yaroslavtsev et al., 2020) | $0.5$ | $O(nK_B)$ |
| | 1-guess-Greedy$^+$ (Feldman et al., 2023) | $0.6174$ | $O(n^2 K_B)$ |
| **Evolutionary algorithms** | POMC (Qian et al., 2017a) | $(1/2)(1 - 1/e)$ | $O(nBP_{max}/\delta)$ |
| | EAMC (Bian et al., 2020) | $(1/2)(1 - 1/e)$ | $O(n^2 K_B)$ |
| | FPOMC (Bian et al., 2021) | $(1/2)(1 - 1/e)$ | $O(n^2 K_B)$ |
| | EVO-SMC (Zhu et al., 2024) | $0.5$ | $O(n^2 K_B)$ |
| | POMC (this paper) | **0.5** | $O(nBP_{max}/\delta)$ |
| | EPOL (this paper) | **0.6174** | $O(n^2 BP_{max}/\delta)$ |

Note that we represent a set $\{v\}$ with a single element as $v$.

Our studied problem as presented in Definition 2.1 is to maximize a monotone objective function $f$ with a linear cost function $c$. We assume w.l.o.g. that monotone functions are normalized, i.e., $f(\emptyset) = 0$.

**Definition 2.1** (Subset Selection with a Linear Cost Constraint). Given a monotone submodular objective function $f : 2^V \to \mathbb{R}^+$, a linear cost function $c : 2^V \to \mathbb{R}^+$ and a budget $B$, to find

$$\arg\max_{X \subseteq V} f(X) \quad \text{s.t.} \quad c(X) = \sum_{v \in X} c(v) \le B. \quad (4)$$

### 2.1. Previous Algorithms

We now introduce practical greedy algorithms for the problem defined in Definition 2.1, including 1-guess-Greedy$^+$ (Feldman et al., 2023), which achieves the best-known practical approximation ratio of 0.6174, close to the optimal ratio of $1 - 1/e \approx 0.632$. In addition, we introduce POMC, an EA that has demonstrated advantages in experiments and achieves an approximation ratio of $(1/2)(1 - 1/e) \approx 0.32$.

#### 2.1.1. GREEDY ALGORITHMS

Generalized Greedy Algorithm (GGA) (Zhang & Vorobeychik, 2016) iteratively selects an item $v$ that maximizes the ratio of marginal gain on $f$ to cost $c$, until reaching the cost budget $B$. The algorithm finally outputs the better of two candidates: the subset $X_i$ found by the greedy process or the best single item. GGA achieves a $(1/2)(1 - 1/e) \approx 0.32$-approximation ratio (Krause & Guestrin, 2005), which was later improved to $1 - 1/\sqrt{e} \approx 0.39$ (Lin & Bilmes, 2010). An alternative greedy algorithm, called Greedy$^+$ (Yaroslavtsev et al., 2020), extends GGA by considering solutions formed by adding a single item to all generated subsets $X_j$ in the iterative process. Specifically, Greedy$^+$ outputs the best solution in $\{X_i\} \cup \{X_j \cup v \mid 0 \le j \le i, v \in V \wedge c(X_j \cup v) \le B\}$. This modification improves the approximation ratio

to 0.5 (Yaroslavtsev et al., 2020). Recently, a practical greedy algorithm named 1-guess-Greedy$^+$ was proposed. By performing a single partial enumeration on the existing Greedy$^+$ algorithm, it achieves an approximation ratio of 0.6174, which is impressively close to the optimal approximation ratio of $1 - 1/e \approx 0.632$ (Feldman et al., 2023).

#### 2.1.2. POMC ALGORITHM

The performance of greedy algorithms may be limited due to the greedy nature. This motivates the design of a series of EAs (Qian et al., 2017a; Bian et al., 2020; 2021; Roostapour et al., 2022; Neumann & Witt, 2023; Neumann & Rudolph, 2024; Zhu et al., 2024), trying to obtain better optimization abilities for subset selection. Based on Pareto Optimization (Friedrich & Neumann, 2015; Qian et al., 2015b), Qian et al. (2017a) proposed a multi-objective EA, POMC, for subset selection. It represents a subset $X \subseteq V$ by a Boolean vector $\boldsymbol{x} \in \{0,1\}^n$, where the $i$-th bit $x_i = 1$ iff $v_i \in X$. We will not distinguish $\boldsymbol{x} \in \{0,1\}^n$ and its corresponding subset $\{v_i \in V \mid x_i = 1\}$ for notational convenience. POMC reformulates the original problem Eq. (4) as a bi-objective maximization problem

$$\arg\max_{\boldsymbol{x} \in \{0,1\}^n} \left( f_1(\boldsymbol{x}), \; f_2(\boldsymbol{x}) \right), \quad (5)$$

where $f_1(\boldsymbol{x}) = \begin{cases} -\infty, & c(\boldsymbol{x}) > B \\ f(\boldsymbol{x}), & \text{otherwise} \end{cases}$, $\quad f_2(\boldsymbol{x}) = -c(\boldsymbol{x})$.

That is, POMC maximizes the objective function $f$ and minimizes the cost function $c$ simultaneously. The solutions with cost values larger than $B$ (i.e., $c(\boldsymbol{x}) > B$) are excluded by setting their $f_1$ values to $-\infty$. Under the bi-objective formulation, two solutions are compared based on the domination relationship.

**Definition 2.2** (Domination). For two solutions $\boldsymbol{x}$ and $\boldsymbol{x}'$,
- $\boldsymbol{x}$ *weakly dominates* $\boldsymbol{x}'$, denoted as $\boldsymbol{x} \succeq \boldsymbol{x}'$, if $f_1(\boldsymbol{x}) \ge f_1(\boldsymbol{x}') \wedge f_2(\boldsymbol{x}) \ge f_2(\boldsymbol{x}')$;
- $\boldsymbol{x}$ *dominates* $\boldsymbol{x}'$, denoted as $\boldsymbol{x} \succ \boldsymbol{x}'$, if $\boldsymbol{x} \succeq \boldsymbol{x}'$ and either

**Algorithm 1** POMC ($V$, $f$, $c$, $B$)

---

**Input**: a ground set $V$ with $n$ items, a monotone submodular function $f$, a linear cost function $c$, a budget $B$

**Output**: a solution $\boldsymbol{x} \in \{0,1\}^n$ with $c(\boldsymbol{x}) \leq B$

**Process**:

1: Reformulate the original problem in Eq. (4) to the bi-objective problem $(f_1(\boldsymbol{x}), f_2(\boldsymbol{x}))$ in Eq. (5);
2: Let $\boldsymbol{x} = 0^n$, $P = \{\boldsymbol{x}\}$;
3: **repeat**
4:     Select $\boldsymbol{x}$ from $P$ uniformly at random;
5:     Generate $\boldsymbol{x}'$ by flipping each bit of $\boldsymbol{x}$ with prob. $1/n$;
6:     **if** $\nexists \boldsymbol{z} \in P$ such that $\boldsymbol{z} \succ \boldsymbol{x}'$ **then**
7:         $P = (P \setminus \{\boldsymbol{z} \in P \mid \boldsymbol{x}' \succeq \boldsymbol{z}\}) \cup \{\boldsymbol{x}'\}$
8:     **end if**
9: **until** some criterion is met
10: **return** $\arg\max_{\boldsymbol{x} \in P, c(\boldsymbol{x}) \leq B} f(\boldsymbol{x})$

---

$f_1(\boldsymbol{x}) > f_1(\boldsymbol{x}')$ or $f_2(\boldsymbol{x}) > f_2(\boldsymbol{x}')$;
• they are *incomparable* if neither $\boldsymbol{x} \succeq \boldsymbol{x}'$ nor $\boldsymbol{x}' \succeq \boldsymbol{x}$.

To solve the bi-objective maximization problem Eq. (5), POMC employs a simple multi-objective EA in lines 2–9 of Algorithm 1, inspired by the GSEMO algorithm (Laumanns et al., 2004). It starts from the empty set $0^n$ (line 2), and repeatedly improves the quality of solutions in the population $P$ (lines 3–9). In each iteration, a parent solution $\boldsymbol{x}$ is selected from $P$ uniformly at random (line 4); then an offspring solution $\boldsymbol{x}'$ is generated by flipping each bit of $\boldsymbol{x}$ with probability $1/n$ (line 5), which is used to update the population $P$ (line 6–8). If $\boldsymbol{x}'$ is not dominated by any solution in $P$ (line 6), it will be added into $P$, and meanwhile, those solutions weakly dominated by $\boldsymbol{x}'$ will be deleted (line 7). This ensures that $P$ contains only incomparable solutions. After terminated, it returns the best feasible solution with the largest $f$ value in the population (line 10).

Qian et al. (2017a) proved that POMC achieves an approximation ratio of $(1/2)(1 - 1/e)$ with at most $enBP_{max}/\delta$ expected number of iterations, where $P_{max}$ is the largest population size during the running of POMC, and $\delta = \min_{v \in V} c(v)$ is the minimum item cost in the ground set $V$.

**Theorem 2.3.** *(Qian et al., 2017a) For the problem in Definition 2.1, POMC with at most $enBP_{max}/\delta$ expected number of iterations finds a subset $X \subseteq V$ such that $c(X) \leq B$ and*

$$f(X) \geq (1/2)(1 - 1/e) \cdot f(X^*),$$

*where $X^*$ is an optimal solution.*

Though the approximation ratio has gaps compared to other algorithms, POMC achieves the best emprical performance (Roostapour et al., 2022). This work aims to improve the approximation bound of POMC, and design a more advanced EA with stronger theoretical guarantees and

better empirical performance, as stated in Sections 3 and 4, respectively. We will show their superior performance on two real-world applications in Section 5.

## 3. Improved $1/2$-Approximation of POMC

In this section, we re-analyze the approximation ratio of POMC, improving the previously known $(1/2)(1 - 1/e)$ in Theorem 2.3 to $1/2$ in Theorem 3.1. Let $o_c$ be the item from the optimal solution $X^*$ with the maximum cost, i.e., $o_c \in \arg\max_{v \in X^*} c(v)$.

**Theorem 3.1.** *For the problem in Definition 2.1, POMC with at most $enBP_{max}/\delta$ expected number of iterations finds a subset $X \subseteq V$ such that $c(X) \leq B$ and*

$$f(X) \geq (1/2) \cdot f(X^*).$$

Previous analysis of POMC used a coarse-grained manner to evaluate the lower bound on improving $f(X)$ (Qian et al., 2017a). This led to POMC being able to derive a solution $X_1$ that satisfies $f(X_1) \geq (1 - 1/e) \cdot f(X^*)$, which is, however, infeasible, i.e., $c(X_1) > B$. A connection was then established between $X_1$ and two feasible solutions $X$ and $Y$, demonstrating that $\max\{f(X), f(Y)\} \geq f(X_1)/2 \geq (1/2)(1 - 1/e) \cdot f(X^*)$. This weakens the tightness of the bound. In contrast, our analysis adopts a fine-grained approach, inspired by the analysis of Greedy+ (Yaroslavtsev et al., 2020) and 1-Guess-Greedy+ (Feldman et al., 2023), to evaluate the lower bound on improving $f(X)$, leading to that POMC can find a feasible solution $X_2$ with $f(X_2) \geq (1/2) \cdot f(X^*)$ and $c(X_2) \in (B - c(o_c), B]$. Then, we give the detailed proof of Theorem 3.1, which relies on Lemma 3.2, showing that POMC can obtain an approximation ratio of $1 - z(r)$ within at most $enBP_{max}/\delta$ expected number of iterations. This approximation ratio, i.e., $1 - z(r)$, based on a special function $z(\cdot)$ described in Lemma 3.4, where $r = f(o_c)/f(X^*)$, will also be used in the analysis of the proposed EPOL algorithm in Section 4.

**Lemma 3.2.** *Define $r$ as $f(o_c)/f(X^*)$. If $r \leq 1/2$, POMC with at most $enBP_{max}/\delta$ expected number of iterations finds a subset $X \subseteq V$ with $c(X) \leq B$ and*

$$f(X) \geq (1 - z(r)) \cdot f(X^*),$$

*where $z(\cdot)$ is a special function described in Lemma 3.4.*

Lemma 3.2 builds on Lemmas 3.3 and 3.4. Specifically, Lemma 3.3 shows a lower bound on improving $f(X)$ by adding a specific item to $X$. The improvement is expressed as a weighted combination of $f(X^*) - f(X \cup o_c)$ and $f(X^*) - f(X) - f(o_c)$, scaled by the cost $c(v^*)$ and normalized by the budget $B - c(o_c)$. Its proof is provided in Appendix A due to space limitation. Lemma 3.4 establishes an one-to-one correspondence between the expressions $r/z(r) - 1$ and $\ln(z(r)/(1 - r))$, as defined by the unique value $z(r)$.

**Lemma 3.3.** *For any subset $X \subseteq V$ such that the total cost $c(X) \leq B - c(o_c)$, where $o_c \in \arg\max_{v \in X^*} c(v)$, there exists a solution $X' = X \cup v^*$ that satisfies*

$$f(X') - f(X) \geq \frac{(1-\alpha) \cdot c(v^*)}{B - c(o_c)} \left( f(X^*) - f(X \cup o_c) \right)$$
$$+ \frac{\alpha \cdot c(v^*)}{B - c(o_c)} \left( f(X^*) - f(o_c) - f(X) \right),$$

*where $v^* \in \arg\max_{v \in X^* \backslash (X \cup o_c)} \frac{f(X \cup v) - f(X)}{c(v)}$ and $\alpha \in [0, 1]$.*

**Lemma 3.4** (Lemma 4 in (Feldman et al., 2023)). *For every $r \in [0, 1/2]$, there is a unique value $z(r) \in [r, 1/2]$ satisfying the equation $r/z(r) - 1 = \ln(z(r)/(1-r)) \in [-1, 0]$. Moreover, $z(r)$ is a non-decreasing function of $r$.*

The proof of Lemma 3.2 defines $J_{max}$ as the largest $j$ in $[0, B - c(o_c)]$ for which a subset $X \in P$ satisfying $c(X) \leq j$ and $f(X)$ meets a threshold that increases with $j$. The process iteratively updates the value of $J_{max}$ by applying Lemma 3.3 until a subset achieves the desired approximation level. Additionally, it analyzes the expected number of iterations required to achieve each successive improvement in $J_{max}$.

**Proof of Lemma 3.2.** We analyze the increase of a quantity $J_{max}$ during the process of POMC, which is defined as

$$J_{max} = \max\{j \in [0, B - c(o_c)] \mid \exists X \in P, c(X) \leq j \wedge$$
$$f(X) \geq (1 - e^{-\frac{\min\{j,J\}}{B - c(o_c)}}) \cdot (f(X^*) - f(o_c))$$
$$+ \frac{\max\{j - J, 0\}}{B - c(o_c)} \cdot z(r) \cdot f(X^*)\},$$

where $r = f(o_c)/f(X^*) \leq 1/2$, and $J = -(B - c(o_c)) \cdot \ln \frac{z(r)}{1-r} \in [0, B - c(o_c)]$ according to Lemma 3.4.

The initial value of $J_{max}$ is 0, since POMC starts from the all-0s solution $0^n$, representing an empty set. Assume that currently $J_{max} = i \leq B - c(o_c)$ and $X$ is a corresponding solution with the value $i$, i.e., $c(X) \leq i \leq B - c(o_c)$ and

$$f(X) \geq (1 - e^{-\frac{\min\{i,J\}}{B - c(o_c)}}) \cdot (f(X^*) - f(o_c))$$
$$+ \frac{\max\{i - J, 0\}}{B - c(o_c)} \cdot z(r) \cdot f(X^*). \tag{6}$$

We first show that $J_{max}$ cannot decrease. If $X$ is kept in $P$, $J_{max}$ obviously will not decrease. If $X$ is deleted from $P$ (line 7 of Algorithm 1), the newly included solution $X'$ must weakly dominate $X$, i.e., $f(X') \geq f(X)$ and $c(X') \leq c(X)$, implying $J_{max} \geq i$.

Next, we show that within $enP_{\max}$ expected number of iterations, either $J_{max}$ can increase by at least $\delta$ or a $(1 - z(r))$-approximation solution can be generated, where $\delta = \min_{v \in V} c(v)$ and $P_{max}$ is the largest size of the population $P$ during the execution of POMC.

**Case 1:** Assume that $f(X \cup o_c) < (1 - z(r)) \cdot f(X^*)$. Because $J_{max} \leq B - c(o_c)$, the corresponding solution $X$ must satisfy $c(X) \leq B - c(o_c)$. According to Lemma 3.3, by flipping one specific 0-bit of $X$, i.e., adding an item $v^* \in \arg\max_{v \in X^* \backslash \{X \cup o_c\}} \frac{f(X \cup v) - f(X)}{c(v)}$, we can generate a new solution $X' = X \cup v^*$ such that

$$f(X') - f(X) \geq \frac{(1-\alpha) \cdot c(v^*)}{B - c(o_c)} \left( f(X^*) - f(X \cup o_c) \right)$$
$$+ \frac{\alpha \cdot c(v^*)}{B - c(o_c)} \left( f(X^*) - f(o_c) - f(X) \right)$$
$$\geq \frac{(1-\alpha) \cdot c(v^*)}{B - c(o_c)} \cdot z(r) \cdot f(X^*) \tag{7}$$
$$+ \frac{\alpha \cdot c(v^*)}{B - c(o_c)} \left( f(X^*) - f(o_c) - f(X) \right),$$

where $\alpha \in [0, 1]$ and the last inequality holds by the assumption that $f(X \cup o_c) < (1 - z(r)) \cdot f(X^*)$.

Eq. (6) consists of two terms added together, where the first term is $(1 - e^{-\frac{\min\{i,J\}}{B - c(o_c)}}) \cdot (f(X^*) - f(o_c))$ and the second term is $\frac{\max\{i - J, 0\}}{B - c(o_c)} \cdot z(r) \cdot f(X^*)$. When $i \leq J$, the second term in Eq. (6) equals zero. Conversely, when $i > J$, the first term becomes a constant value, $(1 - e^{-\frac{J}{B - c(o_c)}}) \cdot (f(X^*) - f(o_c))$, while the second term contributes $\frac{i - J}{B - c(o_c)} \cdot z(r) \cdot f(X^*)$. Next, we analyze $f(X')$ by considering the relationships among $i$, $i + c(v^*)$, and $J$.

(1) When $i + c(v^*) \leq J$, applying Eq. (6) to Eq. (7) and setting $\alpha = 1$, we get

$$f(X') \geq \left(1 - \left(1 - \frac{c(v^*)}{B - c(o_c)}\right) \cdot e^{-\frac{i}{B - c(o_c)}}\right)$$
$$\cdot (f(X^*) - f(o_c)) \tag{8}$$
$$\geq \left(1 - e^{-\frac{i + c(v^*)}{B - c(o_c)}}\right) \cdot (f(X^*) - f(o_c)),$$

where the last inequality holds by $1 + x \leq e^x$.

(2) When $J \in [i, i + c(v^*))$, applying Eq. (6) to Eq. (7), and setting $\alpha = (J - i)/c(v^*) < 1$, i.e., $i + \alpha \cdot c(v^*) = J$, we get

$$f(X') \geq \left(1 - e^{-\frac{i + \alpha \cdot c(v^*)}{B - c(o_c)}}\right) \cdot (f(X^*) - f(o_c))$$
$$+ \frac{(1-\alpha) \cdot c(v^*)}{B - c(o_c)} z(r) \cdot f(X^*) \tag{9}$$
$$= Cst + \frac{i + c(v^*) - J}{B - c(o_c)} \cdot z(r) \cdot f(X^*),$$

where $Cst$ denotes the constant value $Cst = (1 - e^{-\frac{J}{B - c(o_c)}}) \cdot (f(X^*) - f(o_c))$.

(3) When $J < i$, applying Eq. (6) to Eq. (7) and setting $\alpha = 0$, we get

$$f(X') \geq Cst + \frac{i + c(v^*) - J}{B - c(o_c)} \cdot z(r) \cdot f(X^*). \tag{10}$$

By combining the inequalities derived from cases (1) to (3), i.e., Eqs. (8) to (10), we can conclude that the new solution $X' = X \cup v^*$ satisfies:

$$f(X') \geq \left(1 - e^{-\frac{\min\{i + c(v^*), J\}}{B - c(o_c)}}\right) \cdot (f(X^*) - f(o_c))$$
$$+ \frac{\max\{i + c(v^*) - J, 0\}}{B - c(o_c)} \cdot z(r) \cdot f(X^*).$$

Additionally, the cost of solution $X'$ satisfies $c(X') = c(X) + c(v^*) \leq i + c(v^*)$. Consequently, the solution $X'$ must be included in $P$; otherwise, $X'$ must be dominated by one solution in $P$ (line 6 of Algorithm 1). This implies that $J_{max}$ has already exceeded $i$, contradicting the assumption that $J_{max} = i$. Hence, after including $X'$, we have $J_{max} \geq i + c(v^*)$. Since $c(v^*) \geq \delta$, it follows that $J_{max} \geq i + \delta$. Thus, $J_{max}$ can increase by at least $\delta$ in one iteration with probability at least $\frac{1}{P_{max}} \cdot \frac{1}{n}(1 - \frac{1}{n})^{n-1} \geq \frac{1}{enP_{max}}$, where $\frac{1}{P_{max}}$ is a lower bound on the probability of selecting $X$ (line 4), and $\frac{1}{n}(1 - \frac{1}{n})^{n-1}$ is the probability of flipping a specific bit while keeping other bits unchanged (line 5). Therefore, it needs at most $enP_{max}$ expected number of iterations to increase $J_{max}$ by at least $\delta$.

**Case 2:** If $f(X \cup o_c) < (1 - z(r)) \cdot f(X^*)$ does not hold, it implies that $X \cup o_c$ is a feasible solution with $(1 - z(r))$-approximation, i.e., $c(X \cup o_c) \leq i + c(o_c) \leq B$ and

$$f(X \cup o_c) \geq (1 - z(r)) \cdot f(X^*).$$

The solution $X \cup o_c$ can be generated in one iteration by selecting $X$ in line 4 and flipping only the 0-bit corresponding to the item $o_c$ in line 5, whose probability is at least $\frac{1}{P_{max}} \cdot \frac{1}{n}(1 - \frac{1}{n})^{n-1} \geq \frac{1}{enP_{max}}$. That is, $X \cup o_c$ will be generated in at most $enP_{max}$ iterations in expectation. According to the updating procedure of $P$ (lines 6–8), we know that once $X \cup o_c$ is produced, $P$ will always contain a solution $Z \succeq X \cup o_c$, i.e., $c(Z) \leq c(X \cup o_c) \leq B$ and $f(Z) \geq f(X \cup o_c) \geq (1 - z(r)) \cdot f(X^*)$.

We have shown that when $J_{max} \leq B - c(o_c)$, either $J_{max}$ can increase by at least $\delta$ or a $(1 - z(r))$-approximation solution can be generated within $enP_{max}$ iterations in expectation. The latter implies that the theorem already holds. We now focus on the state where the inclusion of the new solution $X'$ causes $J_{max} + c(v^*)$ to exceed $B - c(o_c)$. Specifically, we consider the case where $J_{max} \leq B - c(o_c) < J_{max} + c(v^*)$, which leads to

$$f(X') \geq Cst + \frac{J_{max} + c(v^*) - J}{B - c(o_c)} \cdot z(r) \cdot f(X^*)$$
$$\geq Cst + \frac{B - c(o_c) - J}{B - c(o_c)} \cdot z(r) \cdot f(X^*)$$
$$= \left(1 - \frac{z(r)}{1 - r}\right) \cdot \left(f(X^*) - f(o_c)\right) + r \cdot f(X^*)$$
$$= (1 - z(r)) \cdot f(X^*),$$

where the first inequality holds by applying Eqs. (9) and (10) because $J = -(B - c(o_c)) \cdot \ln \frac{z(r)}{1-r} \leq B - c(o_c) < J_{max} + c(v^*)$, the second inequality holds by $J_{max} + c(v^*) > B - c(o_c)$, the first equality holds by $Cst = (1 - e^{-\frac{J}{B - c(o_c)}}) \cdot (f(X^*) - f(o_c))$, $J = -(B - c(o_c)) \cdot \ln \frac{z(r)}{1-r}$, and $\ln \frac{z(r)}{1-r} = \frac{r}{z(r)} - 1$ by Lemma 3.4, and the last equality is derived from the fact that $r = f(o_c)/f(X^*)$. The solution $X'$ is feasible, because $c(X') \leq J_{max} + c(v^*) \leq B - c(o_c) + c(o_c) = B$. Once $X'$ is produced, $P$ will always contain a $(1 - z(r))$-approximation solution. To achieve this, it requires at most $\frac{B - c(o_c)}{\delta} \cdot enP_{max} + enP_{max} \leq enBP_{max}/\delta$ expected number of iterations. Thus, the lemma holds. $\square$

The proof of Theorem 3.1 proceeds by analyzing two cases. In each case, a $1/2$-approximation solution is achieved. The final result is obtained by taking the maximum of the expected number of iterations required for the two cases.

**Proof of Theorem 3.1.** Lemma 3.2 has shown that when $r = f(o_c)/f(X^*) \leq 1/2$, POMC can find a feasible solution $X$ in the population $P$ satisfying $f(X) \geq (1 - z(r)) \cdot f(X^*)$, using at most $enBP_{max}/\delta$ expected number of iterations. According to Lemma 3.4, the function $z(r)$ has an upper bound $1/2$ when $r \in [0, 1/2]$. Then we have $f(X) \geq (1 - z(r)) \cdot f(X^*) \geq (1/2) \cdot f(X^*)$.

When $f(o_c)/f(X^*) > 1/2$, it implies that the solution with a single item $o_c$ satisfying $f(o_c) > (1/2) \cdot f(X^*)$. Note that $0^n$ always exists in $P$, since it has the smallest cost. Thus, the subset $o_c$ (an abbreviation of $\{o_c\}$) can be generated in one iteration by selecting $0^n$ in line 4 of Algorithm 1 and flipping only the corresponding 0-bit in line 5, whose probability is at least $\frac{1}{P_{max}} \cdot \frac{1}{n}(1 - \frac{1}{n})^{n-1} \geq \frac{1}{enP_{max}}$. That is, $o_c$ will be generated in at most $enP_{max}$ expected number of iterations. According to the updating procedure of $P$ (lines 6-8), we know that once the subset $o_c$ is produced, $P$ will always contain a $1/2$-approximation solution.

**Taking the maximum** of the expected number of iterations of the above two cases, POMC uses at most $enBP_{max}/\delta$ expected number of iterations, to find a $1/2$-approximation solution. Thus, the theorem holds. $\square$

## 4. Proposed EPOL with $0.6174$-Approximation

In this section, we propose an Enhanced Pareto Optimization method for maximizing a monotone submodular function with a Linear cost constraint in Definition 2.1, briefly called EPOL. As described in Algorithm 2, EPOL divides the original problem $(V, f, c, B)$ into several residual problems, that is, for each $v \in V$, it creates a residual problem $(V \setminus v, f(\cdot \mid v), c, B - c(v))$, where $f(\cdot \mid v) = f(\cdot \cup v) - f(v)$, and then solves these residual problems by running POMC (line 3). EPOL combines the output solution $X_v$ of each residual problem $(V \setminus v, f(\cdot \mid v), c, B - c(v))$ with $v$ to

**Algorithm 2** EPOL Algorithm

**Input**: a ground set $V$ with $n$ items, a monotone submodular objective function $f$, a linear cost function $c$, a budget $B$

**Output**: a solution $X \subseteq V$ with $c(X) \leq B$

**Process**:

1: $Q \leftarrow \emptyset$;
2: **for** $v$ in $V$ **do**
3: $\quad X_v \leftarrow \text{POMC}(V \setminus v, f(\cdot \mid v), c, B - c(v))$;
4: $\quad Q \leftarrow Q \cup \{X_v \cup v\}$
5: **end for**
6: **return** $\arg\max_{X \in Q, c(X) \leq B} f(X)$

form a new candidate solution set $Q = \{X_v \cup v, \forall v \in V\}$ (line 4). Finally, EPOL returns the best feasible solution among these candidate solutions (line 6). The idea of EPOL is inspired by (Feldman et al., 2023).

In Theorem 4.1, we prove that EPOL can achieve an approximation ratio of 0.6174 using at most $en^2 B P_{max}/\delta$ expected number of iterations. The key to the proof is establishing a connection between the approximation guarantees of the residual problem $(V \setminus o_f, f(\cdot \mid o_f), c, B - c(o_f))$ and the original problem, primarily by using Lemma 3.2, where $o_f \in \arg\max_{v \in X^*} f(v)$.

**Theorem 4.1.** *For the problem in Definition 2.1, EPOL with at most $en^2 B P_{max}/\delta$ expected number of iterations finds a subset $X \subseteq V$ such that $c(X) \leq B$ and*

$$f(X) \geq 0.6174 \cdot f(X^*).$$

*Proof.* Let $o_f$ be the largest-value item in the optimal solution $X^*$, i.e., $o_f \in \arg\max_{v \in X^*} f(v)$. We can observe that $X^* \setminus o_f$ is an optimal solution of the residual problem $(V \setminus o_f, f(\cdot \mid o_f), c, B - c(o_f))$. For any $X \subseteq V \setminus o_f$ with $c(X) \leq B - c(o_f)$, it holds that

$$f(X^* \setminus o_f \mid o_f) \geq f(X \mid o_f).$$

We now prove Theorem 4.1 by considering the relationship between $f(o_f)$ and $f(X^*)$.

**Case 1:** Assume that $f(o_f) \geq (1/3) \cdot f(X^*)$. Given that the function $f$ is submodular, it can be verified that $f(\cdot \mid v)$ is also submodular. According to Theorem 3.1, for the residual problem $(V \setminus o_f, f(\cdot \mid o_f), c, B - c(o_f))$, POMC will return a solution $X_{o_f}$ satisfying that $X_{o_f} \leq B - c(o_f)$ and

$$f(X_{o_f} \mid o_f) \geq (1/2) \cdot f(X^* \setminus o_f \mid o_f).$$

By rearranging the above inequality, we obtain

$$f(X_{o_f} \cup o_f) \geq f(X^*)/2 + f(o_f)/2 \geq (2/3) \cdot f(X^*),$$

where the last inequality is by the assumption $f(o_f) \geq (1/3) \cdot f(X^*)$. The solution $X_{o_f} \cup o_f$ will be contained in $Q$ in line 4 of Algorithm 2.

**Case 2:** We analyze the case where $f(o_f) < (1/3) \cdot f(X^*)$. Let $o_c'$ be the item in $X^* \setminus o_f$ with the maximum cost, i.e., $o_c' \in \arg\max_{v \in X^* \setminus o_f} c(v)$, which implies $f(o_c') \leq f(o_f)$. By the submodularity (i.e., Eq. (2)) of $f$, we have $f(X^* \setminus o_f) - f(\emptyset) \geq f(X^*) - f(o_f)$. It follows that

$$\frac{f(o_c')}{f(X^* \setminus o_f)} \leq \frac{f(o_f)}{f(X^*) - f(o_f)} \leq 1/2. \quad (11)$$

Let $r' = f(o_c')/f(X^* \setminus o_f) \leq 1/2$. We apply Lemma 3.2 to the residual problem $(V \setminus o_f, f(\cdot \mid o_f), c, B - c(o_f))$. As a result, POMC returns a solution $X_{o_f}$ satisfying $X_{o_f} \leq B - c(o_f)$ and $f(X_{o_f} \mid o_f) \geq (1 - z(r')) \cdot f(X^* \setminus o_f \mid o_f)$. Rearranging this inequality yields

$$f(X_{o_f} \cup o_f) \geq (1 - z(r')) \cdot (f(X^*) - f(o_f)) + f(o_f)$$
$$\geq (1 - z(\frac{f(o_f)}{f(X^*) - f(o_f)})) \cdot (f(X^*) - f(o_f)) + f(o_f),$$

where the second inequality holds by Eq. (11) and the non-decreasing property of $z(\cdot)$. Let $t = f(o_f)/f(X^*)$, where $t \in [0, 1/3]$. Then, the above equation becomes $f(X_{o_f} \cup o_f) \geq \left((1 - z(\frac{t}{1-t})) \cdot (1 - t) + t\right) \cdot f(X^*) \geq 0.6174 \cdot f(X^*)$, where the second inequality holds by Lemma 13 of (Feldman et al., 2023). The solution $X_{o_f} \cup o_f$ will be contained in $Q$ in line 4 of Algorithm 2.

Hence, EPOL guarantees a solution $X$ such that $c(X) \leq B$ and $f(X) \geq \min\{2/3, 0.6174\} \cdot f(X^*) = 0.6174 \cdot f(X^*)$. As EPOL runs POMC $n$ times for solving the $n$ residual problems, the total expected number of iterations is at most $n \cdot enBP_{max}/\delta$. Thus, the theorem holds. $\square$

The processes in lines 2–5 of Algorithm 2 are independent and can run in parallel on $N$ processors. This reduces the expected number of iterations to $en^2 BP_{max}/(N\delta)$ (Corollary 4.2). When $N = n$, the expected number of iterations can reduce to $enBP_{max}/\delta$, as same as that of POMC.

**Corollary 4.2.** *For the problem in Definition 2.1, EPOL with at most $en^2 BP_{max}/(N\delta)$ expected number of iterations finds a subset $X \subseteq V$ such that $c(X) \leq B$ and $f(X) \geq 0.6174 \cdot f(X^*)$, where $N$ denotes the number of processors used by EPOL.*

Note that in our experiments, EPOL executes only on $K_B$ residual problems corresponding to the top-$K_B$ items with the highest $f$ values, where $K_B$ denotes the maximum number of items in a subset $X$ that satisfies the budget constraint $c(X) \leq B$. Notably, $K_B$ is non-decreasing with respect to the budget $B$, implying that EPOL will execute more residual problems as the problem becomes harder (i.e., when the budget $B$ increases). Despite this limitation, EPOL still shows superior performance.

# 5. Empirical Study

We empirically examine EPOL on the applications of maximum coverage and influence maximization, by comparing a series of competitive algorithms, including greedy algorithms and the EA-based methods. The source code is available at https://github.com/lamda-bbo/EPOL.

## 5.1. Experimental Settings

**Maximum Coverage.** Given a set $U$ of items, and a collection $V = \{S_1, S_2, \ldots, S_n\}$ of subsets of $U$, maximum coverage (Feige, 1998) is to select a subset of $V$ to maximize the number of covered items of $U$ under a cost budget $B$, i.e., $\arg\max_{X \subseteq V, c(X) \leq B} \left| \bigcup_{S_i \in X} S_i \right|$. The objective function is easily verified to be monotone and submodular. We use three real-world graph datasets: *frb30-15-1* (450 vertices, 17,827 edges) and *frb35-17-1* (595 vertices, 27,856 edges), both studied in (Bian et al., 2020; Roostapour et al., 2022), as well as the Twitter Interaction Network for the US Congress, *congress* (475 vertices, 13,289 edges) (Fink et al., 2023). For each vertex, we generate a set which contains the vertex itself and its adjacent vertices. The cost of each vertex (set) is $c(v) = 1 + \max\{d(v) - q, 0\}$ as in (Harshaw et al., 2019), where $d(v)$ is the out-degree of $v$ and $q$ is a constant. We set $q \in \{0, 5, 10\}$ and the budget $B \in \{300, 350, \ldots, 500\}$.

**Influence Maximization.** Given a directed graph $G = (V, E)$ representing a social network, influence maximization (Kempe et al., 2003) is to find a subset of users $X \subseteq V$ such that the expected number of users activated by propagating from $X$ is maximized, while satisfying the cost constraint, i.e., $\arg\max_{X \subseteq V, c(X) \leq B} \mathbb{E}[|IC(X)|]$. $IC(X)$ is the set of users activated by propagating from the seed users $X$ under the Independence Cascade model. It begins with $X$, uses a set $A_t$ to record the nodes activated at time $t$, and at time $t + 1$, each inactive neighbor $v$ of $u \in A_t$ becomes active with edge probability $p_{u,v}$; this process is repeated until no nodes get activated at some time. The function $\mathbb{E}[|IC(X)|]$ is monotone and submodular. We use three real-world graph datasets: *graph100* (100 vertices, 3,465 edges) and *graph200* (200 vertices, 9,950 edges), widely used as social networks in influence maximization (Bian et al., 2020), as well as the animal interaction network *insecta* (152 vertices, 6,716 edges). The cost of each node is calculated based on its out-degree $d(v)$, i.e., $c(v) = 1 + (1 + |\xi|) \cdot d(v)$, where $\xi$ is a random number drawn from the normal distribution $\mathcal{N}(0, 0.5^2)$. To calculate $\mathbb{E}[|IC(X)|]$ in our experiments, we simulate the random propagation process starting from the solution $X$ for 500 times independently, and use the average as an estimation. We set the probability of each edge as $\{0.05, 0.1\}$ and the budget $B$ as $\{100, 200, \ldots, 500\}$.

**Settings.** We compare the SOTA greedy algorithms and

EA-based methods as follows:

- GGA (Zhang & Vorobeychik, 2016): Iteratively selects an item maximizing the ratio of the increment on $f$ and $c$, and outputs the best solution among the subset $X_i$ found by the iterative process and single items, i.e., $\{X_i\} \cup \{v \mid v \in V\}$.

- Greedy$^+$ (Yaroslavtsev et al., 2020): Extends GGA by outputting the best solution in $\{X_i\} \cup \{X_j \cup v \mid 0 \leq j \leq i, v \in V \wedge c(X_j \cup v) \leq B\}$.

- 1-guess-Greedy$^+$ (Feldman et al., 2023): Performs a single partial enumeration on Greedy$^+$.

- POMC (Qian et al., 2017a): Maximizes $f$ and minimizes $c$ simultaneously, through a multi-objective EA process.

- EAMC (Bian et al., 2020): Searches solutions guided by a surrogate function of integrating $f$ and $c$, and maintains a limited number of solutions for each possible subset size.

- EVO-SMC (Zhu et al., 2024): Similar to EAMC, but uses a different surrogate function of integrating function $f$ and function $c$.

- FPOMC (Bian et al., 2021): Modifies POMC by introducing a greedy selection strategy.

The algorithms in our study fall into two categories: Fixed-time algorithms such as GGA, Greedy$^+$, and 1-guess Greedy$^+$, with runtime complexities of $O(nK_B)$, $O(nK_B)$, and $O(n^2 K_B)$, respectively, and anytime algorithms such as POMC, EAMC, FPOMC, and EVO-SMC, whose performance improves with runtime. To ensure a fair comparison, the number of objective evaluations for POMC, EAMC, FPOMC, and EVO-SMC is set to $20nK_B$. EPOL divides the original problem into $n$ residual problems $(V \setminus v, f(\cdot \mid v), c, B - c(v))$ for each $v \in V$. Note that we only run the residual problems corresponding to the top-$K_B$ values of $f(v)$ to balance computational efficiency and performance, rather than all $n$ residual problems. These $K_B$ subproblems are then solved by POMC in parallel with $20nK_B$ evaluations. The ratios of $K_B/n$ in our experimental settings can be found in Table 4 of Appendix B.2. For all EA-based methods, we independently repeat the run 10 times and report the average results. The objective evaluation for influence maximization, i.e., $E[|IC(X)|]$, is noisy because the propagation process is randomized, and we use the average of multiple Monte Carlo simulations to estimate the expectation. Specifically, starting from a solution $X$, we simulate the propagation 500 times and use the average as the estimated objective value. Since the behavior of greedy algorithms is randomized under noise, we also repeat their

*Table 2.* The objective value (number of covered vertices) of maximum coverage (avg $\pm$ std) obtained by the algorithms on *frb35-17-1* for $q = 5$ and the budgets $B \in \{300, 350, \dots, 500\}$. For each $B$, the largest number is bolded, and '$\bullet/\circ$' denote that EPOL is significantly better/worse than the corresponding algorithm by the Wilcoxon signed-rank test with confidence level 0.05. Avg.R. denotes the average rank (the smaller, the better) of each algorithm under each setting as in (Demsar, 2006).

| Budget $B$ | 300 | 350 | 400 | 450 | 500 | Avg.R. |
|---|---|---|---|---|---|---|
| GGA (Zhang & Vorobeychik, 2016) | 309.0 $\bullet$ | 339.0 $\bullet$ | 368.0 $\bullet$ | 401.0 $\bullet$ | 424.0 $\bullet$ | 5.4 |
| Greedy$^+$ (Yaroslavtsev et al., 2020) | 309.0 $\bullet$ | 339.0 $\bullet$ | 368.0 $\bullet$ | 401.0 $\bullet$ | 425.0 $\bullet$ | 4.4 |
| 1-guess- Greedy$^+$ (Feldman et al., 2023) | 314.0 $\bullet$ | 352.0 $\bullet$ | 385.0 $\bullet$ | 412.0 $\bullet$ | 440.0 $\bullet$ | 2.6 |
| EVO-SMC (Zhu et al., 2024) | 292.6 $\pm$ 6.2 $\bullet$ | 325.8 $\pm$ 5.1 $\bullet$ | 355.3 $\pm$ 5.7 $\bullet$ | 381.9 $\pm$ 6.0 $\bullet$ | 410.5 $\pm$ 4.4 $\bullet$ | 8.0 |
| FPOMC (Bian et al., 2021) | 303.9 $\pm$ 4.3 $\bullet$ | 336.1 $\pm$ 3.9 $\bullet$ | 368.6 $\pm$ 4.4 $\bullet$ | 398.1 $\pm$ 3.9 $\bullet$ | 423.2 $\pm$ 5.4 $\bullet$ | 6.0 |
| EAMC (Bian et al., 2020) | 297.1 $\pm$ 3.2 $\bullet$ | 333.5 $\pm$ 5.0 $\bullet$ | 364.1 $\pm$ 5.1 $\bullet$ | 398.5 $\pm$ 4.1 $\bullet$ | 425.5 $\pm$ 5.7 $\bullet$ | 6.2 |
| POMC (Qian et al., 2017a) | 314.6 $\pm$ 1.8 $\bullet$ | 350.9 $\pm$ 2.4 $\bullet$ | 383.7 $\pm$ 2.6 $\bullet$ | 413.8 $\pm$ 2.6 $\bullet$ | 441.5 $\pm$ 2.4 $\bullet$ | 2.4 |
| EPOL (this paper) | **319.1 $\pm$ 0.8** | **356.6 $\pm$ 0.9** | **389.8 $\pm$ 0.6** | **419.0 $\pm$ 0.6** | **445.7 $\pm$ 0.6** | 1.0 |

*Table 3.* Average ranks of algorithms on datasets: MC1 (*frb30-15-1*), MC2 (*frb-35-17-1*), MC3 (*congress*) for maximum coverage with $q = 5$; IM1 (*graph100*), IM2 (*graph200*), IM3 (*insecta*) for influence maximization with an edge probability of 0.05. The smaller, the better.

| | MC1 | MC2 | MC3 | IM1 | IM2 | IM3 |
|---|---|---|---|---|---|---|
| GGA | 5.4 | 5.4 | 6.0 | 7.8 | 8.0 | 8.0 |
| Greedy$^+$ | 4.0 | 4.4 | 5.0 | 6.4 | 7.0 | 6.6 |
| 1-guess-Greedy$^+$ | 2.4 | 2.6 | 2.2 | 4.0 | 5.4 | 3.6 |
| EVO-SMC | 8.0 | 8.0 | 7.0 | 5.6 | 3.6 | 4.8 |
| FPOMC | 7.0 | 6.0 | 8.0 | 5.4 | 5.2 | 5.0 |
| EAMC | 5.6 | 6.2 | 4.0 | 3.0 | 2.2 | 3.4 |
| POMC | 2.6 | 2.4 | 2.8 | 2.8 | 3.0 | 3.2 |
| EPOL | 1.0 | 1.0 | 1.0 | 1.0 | 1.6 | 1.4 |

run 10 times independently and report the average results for the application of influence maximization.

### 5.2. Experimental Results

We report the average results of maximum coverage obtained by each algorithm on *frb35-17-1* with $q = 5$ in Table 2. Additional results are provided in Tables 5 to 9 of Appendix B.2 due to space limitation. Across all settings, EPOL remains significantly superior in almost all cases and is never significantly outperformed.

We summarize the average rank on two applications in Table 3. Among the three greedy methods, GGA performs the worst, while 1-guess-Greedy$^+$ performs the best. For EA-based methods, EAMC, FPOMC, and EVO-SMC underperform greedy methods in maximum coverage but can excel in influence maximization, indicating performance instability across tasks. Overall, POMC is the best performer except EPOL, showing a small and robust average rank across both applications. EPOL clearly performs the best. We also compare Sto-EVO-SMC (Zhu et al., 2024), a stochastic variant of EVO-SMC, which shows minimal difference across parameter settings and underperforms compared to EPOL (Figure 1, Appendix B.3).

To further assess the effectiveness of EPOL, we compare it with P-POMC, a variant that solves the original problem $K_B$ times independently using $K_B$ parallel processors instead of $K_B$ independent residual problems, where each processor is also allocated a budget of $20nK_B$ evaluations. The best solution among $K_B$ processors is selected as the final output for a single run. We run P-POMC 10 times and compare it with EPOL. EPOL consistently achieves better solutions (Table 10, Appendix B.4). Although the setting that EPOL runs a subset of residual problems ($K_B$ residual problems) is sufficient to show the superiority of the proposed EPOL as the implemented version is weaker, we conduct additional experiments comparing EPOL (as in the previous experiments) with the full version (EPOL-full), which enumerates all residual problems. The results of Table 11 in Appendix B.5 show that EPOL-full consistently outperforms EPOL with significant advantages in several cases, highlighting EPOL-full's potential to improve performance by addressing all residual problems. Figures 2 and 3 in Appendix B.6 illustrate the curves of the average values over runtime, where EPOL surpasses all greedy algorithms within $12nK_B$ and $4nK_B$ runtime for maximum coverage and influence maximization, respectively.

## 6. Conclusion

In this paper, we improve the approximation ratio of POMC to $1/2$, providing a clearer theoretical understanding for its promising empirical results. Additionally, we propose EPOL, a novel multi-objective EA that achieves the best-known practical approximation guarantee of $0.6174$ while demonstrating superior empirical performance. We believe this work can advance solving the subset selection problem but also deepen our theoretical understanding of EAs.

## Impact Statement

Subset selection with a linear cost constraint is a fundamental problem in many applications. While EAs have shown competitive empirical performance, they lag behind greedy algorithms in approximation guarantees. This paper

deepens theoretical understanding of EAs and advances the solving of subset selection. We refine the analysis of the SOTA EA, POMC, improving its approximation guarantee to $1/2$, and introduce a novel EA, EPOL, which achieves the best-known practical approximation ratio of 0.6174 and demonstrates superior empirical performance.

## Acknowledgements

This work was supported by the National Science and Technology Major Project (2022ZD0116600), the National Science Foundation of China (62276124), and the Fundamental Research Funds for the Central Universities (14380020).

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

## A. Omitted Proof of Lemma 3.3

**Lemma A.1.** *For any subset $X \subseteq V$ such that the total cost $c(X) \leq B - c(o_c)$, where $o_c \in \arg\max_{v \in X^*} c(v)$, there exists a solution $X' = X \cup v^*$ that satisfies*

$$f(X') - f(X) \geq \frac{(1-\alpha) \cdot c(v^*)}{B - c(o_c)} \left( f(X^*) - f(X \cup o_c) \right) + \frac{\alpha \cdot c(v^*)}{B - c(o_c)} \left( f(X^*) - f(o_c) - f(X) \right),$$

*where $v^* \in \underset{v \in X^* \setminus (X \cup o_c)}{\arg\max} \frac{f(X \cup v) - f(X)}{c(v)}$ and $\alpha \in [0,1]$.*

*Proof.* By the definition of submodularity, we have

$$
\begin{aligned}
f(X^* \cup X \cup o_c) - f(X \cup o_c) &\leq \sum_{v \in X^* \setminus (X \cup o_c)} f(X \cup o_c \cup v) - f(X \cup o_c) \\
&\leq \sum_{v \in X^* \setminus (X \cup o_c)} f(X \cup v) - f(X) = \sum_{v \in X^* \setminus (X \cup o_c)} c(v) \cdot \frac{f(X \cup v) - f(X)}{c(v)} \qquad (12) \\
&\leq \frac{f(X \cup v^*) - f(X)}{c(v^*)} \cdot \sum_{v \in X^* \setminus (X \cup o_c)} c(v) \leq \frac{B - c(o_c)}{c(v^*)} \cdot \left( f(X \cup v^*) - f(X) \right),
\end{aligned}
$$

where the first inequality holds by Eq. (3), the second inequality holds by Eq. (2), the third inequality holds by the definition of $v^*$, and the last inequality holds because the total cost of items in $X^* \setminus (X \cup o_c)$ does not exceed $B - c(o_c)$.

Since $f$ is monotone, we also have $f(X^* \cup X \cup o_c) \geq f(X^*)$. Combining this with Eq. (12), we have

$$
\begin{aligned}
f(X \cup v^*) - f(X) &\geq \frac{c(v^*)}{B - c(o_c)} (f(X^*) - f(X \cup o_c)) \\
&\geq \frac{(1-\alpha) \cdot c(v^*)}{B - c(o_c)} (f(X^*) - f(X \cup o_c)) + \frac{\alpha \cdot c(v^*)}{B - c(o_c)} \left( f(X^*) - f(o_c) - f(X) \right),
\end{aligned}
$$

where $\alpha \in [0,1]$, and the second inequality is by the submodularity (i.e., Eq. (2)) of $f$, that is, $f(X \cup o_c) - f(X) \leq f(o_c) - f(\emptyset) = f(o_c)$. □

## B. Additional Experimental Results

### B.1. Settings

For the application of maximum coverage, we use three real-world graph datasets: *frb30-15-1* (450 vertices, 17,827 edges), *frb35-17-1* (595 vertices, 27,856 edges), and *congress* (475 vertices, 13,289 edges) (Fink et al., 2023). We set $q \in \{0, 5, 10\}$ and the budget $B \in \{300, 350, \ldots, 500\}$. For the application of ifluence maximization, we use three real-world graph datasets: *graph100* (100 vertices, 3,465 edges), *graph200* (200 vertices, 9,950 edges), and *insecta* (152 vertices, 6,716 edges). We set the probability of each edge as $\{0.05, 0.1\}$ and the budget $B$ as $\{100, 200, \ldots, 500\}$.

We compare EPOL with several competitive algorithms, including SOTA greedy algorithms such as GGA, Greedy$^+$ and 1-guess-Greedy+, and the EA-based methods such as POMC, EAMC, FPOMC and EVO-SMC. The number of objective evaluations for POMC, EAMC, FPOMC and EVO-SMC is set to $20nK_B$. EPOL runs the process of POMC in parallel to solve $K_B$ residual problems. We list the ratios of $K_B/n$ in our experimental settings in Table 4. The number of $K_B$ used in our experiments is no more than 32% of $n$.

For all EA-based methods, we independently repeat the run 10 times. Since the behavior of greedy algorithms is randomized under noise, we also repeat their run 10 times independently for the application of influence maximization.

### B.2. Additional Results on All Settings

In this section, we report the average results of each algorithm across various settings. Specifically, we detail the average results on maximum coverage with $q \in \{0, 5, 10\}$ in Tables 5 to 7, respectively. For the application of influence maximization, we report the average results with a probability of each edge as $\{0.05, 0.1\}$ in Tables 8 and 9, respectively.

*Table 4.* Ratios of $K_B/n$ for different settings in two applications. For maximum coverage, $(\cdot, \cdot, \cdot)$ represents $K_B/n$ ratios for $q = \{0, 5, 10\}$. For influence maximization, $(\cdot, \cdot)$ indicates $K_B/n$ ratios at edge probabilities $\{0.05, 0.1\}$.

| | Maximum coverage | | | | |
|---|---|---|---|---|---|
| $B$ | 300 | 350 | 400 | 450 | 500 |
| *frb30-15-1* (450 vertices, 17,827 edges) | (12%, 16%, 21%) | (12%, 17%, 23%) | (13%, 18%, 24%) | (14%, 19%, 25%) | (15%, 20%, 26%) |
| *frb35-17-1* (595 vertices, 27,856 edges) | (2%, 3%, 3%) | (3%, 3%, 4%) | (3%, 3%, 4%) | (3%, 4%, 5%) | (3%, 4%, 5%) |
| *congress* (475 vertices, 13,289 edges) | (7%, 12%, 19%) | (8%, 13%, 21%) | (9%, 14%, 22%) | (10%, 15%, 23%) | (11%, 16%, 25%) |
| | Influence maximization | | | | |
| $B$ | 100 | 200 | 300 | 400 | 500 |
| *graph100* (100 vertices, 3,465 edges) | (14%, 14%) | (20%, 20%) | (25%, 25%) | (29%, 29%) | (32%, 32%) |
| *graph200* (200 vertices, 9,950 edges) | (8%, 8%) | (12%, 12%) | (15%, 15%) | (17%, 17%) | (19%, 19%) |
| *insecta* (152 vertices, 6,716 edges) | (11%, 11%) | (16%, 16%) | (19%, 19%) | (22%, 22%) | (25%, 25%) |

As expected, the objective value achieved by each algorithm increases with $B$ due to the monotonicity of the objective. Among the three greedy methods, GGA consistently performs the worst, Greedy$^+$ performs slightly better than GGA, and 1-guess-Greedy$^+$ performs best, occasionally surpassing EA methods. The performance order of these greedy methods is reasonable because Greedy$^+$ is an extension of GGA, and 1-guess-Greedy$^+$ is further enhanced by performing a single partial enumeration on Greedy$^+$. Therefore, the solutions found by Greedy$^+$ contain those of GGA, and the solutions found by 1-guess-Greedy$^+$ contain those of Greedy$^+$. This relationship explains the progressive improvement in performance.

For EA-based methods, EAMC, FPOMC, and EVO-SMC generally perform worse than greedy methods in most cases when it comes to maximum coverage. However, they tend to excel in influence maximization, as shown by their lower average rank in this task. This suggests that their performance is inconsistent and varies significantly across different tasks. Overall, POMC stands out as the most reliable and well-rounded performer among all existing algorithms, maintaining lower average ranks across various settings in both applications. It performs on par with 1-guess-Greedy$^+$ in maximum coverage, as their average ranks are closely matched, while in influence maximization, POMC achieves a better overall ranking than other EA-based methods. Across all settings, EPOL significantly outperforms other algorithms in nearly all cases, as confirmed by the Wilcoxon signed-rank test (Demsar, 2006) at a 0.05 confidence level. Exceptions are observed in specific cases involving POMC and 1-guess-Greedy$^+$ on maximum coverage, as well as EAMC and EVO-SMC on influence maximization, under certain combinations of the budget $B$. However, EPOL is never significantly outperformed in any scenario.

*Table 5.* The objective value (number of covered vertices) of maximum coverage (avg $\pm$ std) obtained by the algorithms when $q = 0$ and the budgets $B \in \{300, 350, \ldots, 500\}$. For each $B$, the largest number is bolded, and '●/○' denote that EPOL is significantly better/worse than the corresponding algorithm by the Wilcoxon signed-rank test with confidence level 0.05. Avg.R. denotes the average rank (the smaller, the better) of each algorithm under each setting as in (Demsar, 2006).

| | *frb30-15-1* (450 vertices, 17,827 edges) | | | | | |
|---|---|---|---|---|---|---|
| Budget $B$ | 300 | 350 | 400 | 450 | 500 | Avg.R. |
| GGA (Zhang & Vorobeychik, 2016) | 260.0 ● | 291.0 ● | 315.0 ● | 346.0 ● | 358.0 ● | 6.0 |
| Greedy$^+$ (Yaroslavtsev et al., 2020) | 260.0 ● | 291.0 ● | 318.0 ● | 346.0 ● | 364.0 ● | 5.0 |
| 1-guess-Greedy$^+$ (Feldman et al., 2023) | 267.0 ● | 298.0 ● | 326.0 ● | 350.0 ● | 367.0 ● | 2.6 |
| EVO-SMC (Zhu et al., 2024) | 259.5 ± 1.9 ● | 289.5 ± 3.0 ● | 318.1 ± 2.8 ● | 334.9 ± 3.1 ● | 353.9 ± 2.5 ● | 6.8 |
| FPOMC (Bian et al., 2021) | 254.8 ± 5.2 ● | 286.7 ± 5.2 ● | 313.9 ± 3.9 ● | 336.4 ± 7.2 ● | 353.7 ± 3.7 ● | 7.8 |
| EAMC (Bian et al., 2020) | 263.3 ± 2.1 ● | 294.7 ± 3.6 ● | 324.7 ± 3.2 ● | 345.9 ± 2.3 ● | 365.0 ± 2.9 ● | 4.4 |
| POMC (Qian et al., 2017a) | 267.2 ± 1.6 ● | 300.2 ± 1.3 | 325.6 ± 2.1 ● | 349.1 ± 1.5 ● | 369.1 ± 3.1 ● | 2.4 |
| EPOL (this paper) | **269.9 ± 0.9** | **301.4 ± 1.0** | **330.0 ± 0.6** | **352.9 ± 0.3** | **373.1 ± 0.9** | 1.0 |
| | *frb35-17-1* (595 vertices, 27,856 edges) | | | | | |
| GGA (Zhang & Vorobeychik, 2016) | 273.0 ● | 312.0 ● | 346.0 ● | 373.0 ● | 399.0 ● | 6.2 |
| Greedy$^+$ (Yaroslavtsev et al., 2020) | 273.0 ● | 312.0 ● | 346.0 ● | 373.0 ● | 400.0 ● | 5.2 |
| 1-guess-Greedy$^+$ (Feldman et al., 2023) | **283.0** | **321.0** | 353.0 ● | 386.0 ● | 415.0 ● | 2.0 |
| EVO-SMC (Zhu et al., 2024) | 276.0 ± 2.1 ● | 308.5 ± 3.7 ● | 337.6 ± 2.7 ● | 364.4 ± 4.0 ● | 391.2 ± 2.4 ● | 7.4 |
| FPOMC (Bian et al., 2021) | 271.0 ± 2.6 ● | 309.4 ± 3.8 ● | 339.8 ± 2.7 ● | 369.6 ± 5.3 ● | 396.8 ± 5.5 ● | 7.2 |
| EAMC (Bian et al., 2020) | 277.6 ± 2.4 ● | 312.9 ± 3.3 ● | 347.7 ± 2.0 ● | 377.8 ± 3.4 ● | 409.2 ± 1.5 ● | 4.0 |
| POMC (Qian et al., 2017a) | 280.3 ± 2.0 ● | 317.8 ± 1.9 ● | 353.1 ± 1.3 ● | 382.9 ± 2.5 ● | 412.6 ± 3.2 ● | 2.8 |
| EPOL (this paper) | 282.9 ± 0.3 | **321.0 ± 0.0** | **355.2 ± 0.4** | **388.0 ± 0.9** | **417.6 ± 0.5** | 1.2 |
| | *congress* (475 vertices, 13,289 edges) | | | | | |
| GGA (Zhang & Vorobeychik, 2016) | 271.0 ● | 298.0 ● | 320.0 ● | 341.0 ● | 358.0 ● | 6.6 |
| Greedy$^+$ (Yaroslavtsev et al., 2020) | 271.0 ● | 298.0 ● | 322.0 ● | 341.0 ● | 358.0 ● | 5.6 |
| 1-guess-Greedy$^+$ (Feldman et al., 2023) | 282.0 ● | 304.0 ● | 331.0 ● | 355.0 ● | 372.0 ● | 4.0 |
| EVO-SMC (Zhu et al., 2024) | 274.9 ± 7.4 ● | 305.9 ± 6.8 ● | 324.7 ± 7.4 ● | 339.3 ± 3.5 ● | 351.3 ± 4.9 ● | 5.6 |
| FPOMC (Bian et al., 2021) | 248.1 ± 6.8 ● | 275.9 ± 7.6 ● | 299.3 ± 8.9 ● | 319.7 ± 8.6 ● | 337.4 ± 4.5 ● | 8.0 |
| EAMC (Bian et al., 2020) | 277.0 ± 4.3 ● | 307.6 ± 7.5 ● | 332.0 ± 7.8 ● | 358.7 ± 1.0 ● | 372.4 ± 2.2 ● | 3.2 |
| POMC (Qian et al., 2017a) | 282.7 ± 0.5 ● | 316.2 ± 0.9 ● | 339.7 ± 0.6 ● | 359.4 ± 1.4 ● | 376.4 ± 1.1 ● | 2.0 |
| EPOL (this paper) | **283.0 ± 0.0** | **317.0 ± 0.0** | **341.0 ± 0.0** | **360.9 ± 0.3** | **378.0 ± 0.0** | 1.0 |

*Table 6.* The objective value (number of covered vertices) of maximum coverage (avg $\pm$ std) obtained by the algorithms when $q = 5$ and the budgets $B \in \{300, 350, \ldots, 500\}$. For each $B$, the largest number is bolded, and '•/∘' denote that EPOL is significantly better/worse than the corresponding algorithm by the Wilcoxon signed-rank test with confidence level 0.05. Avg.R. denotes the average rank (the smaller, the better) of each algorithm under each setting as in (Demsar, 2006).

| | *frb30-15-1* (450 vertices, 17,827 edges) | | | | | |
|---|---|---|---|---|---|---|
| Budget $B$ | 300 | 350 | 400 | 450 | 500 | Avg.R. |
| GGA (Zhang & Vorobeychik, 2016) | 292.0 • | 318.0 • | 344.0 • | 358.0 • | 377.0 • | 5.4 |
| Greedy$^+$ (Yaroslavtsev et al., 2020) | 292.0 • | 322.0 • | 344.0 • | 361.0 • | 377.0 • | 4.0 |
| 1-guess-Greedy$^+$ (Feldman et al., 2023) | 297.0 • | 324.0 • | 346.0 • | 370.0 • | 386.0 • | 2.4 |
| EVO-SMC (Zhu et al., 2024) | 274.9 ± 4.0 • | 305.6 ± 3.9 • | 330.8 ± 4.3 • | 347.5 ± 5.0 • | 365.0 ± 3.5 • | 8.0 |
| FPOMC (Bian et al., 2021) | 283.2 ± 3.4 • | 311.0 ± 5.0 • | 332.6 ± 3.7 • | 356.2 ± 3.2 • | 371.3 ± 4.2 • | 7.0 |
| EAMC (Bian et al., 2020) | 288.9 ± 3.5 • | 320.7 ± 4.0 • | 340.3 ± 4.1 • | 358.8 ± 4.9 • | 375.2 ± 3.5 • | 5.6 |
| POMC (Qian et al., 2017a) | 295.9 ± 2.1 • | 323.6 ± 1.7 • | 348.4 ± 2.6 • | 369.2 ± 2.6 • | 386.3 ± 2.1 • | 2.6 |
| EPOL (this paper) | **301.1 ± 1.0** | **329.7 ± 0.5** | **354.4 ± 0.9** | **375.2 ± 1.5** | **394.1 ± 1.8** | 1.0 |
| | *frb35-17-1* (595 vertices, 27,856 edges) | | | | | |
| GGA (Zhang & Vorobeychik, 2016) | 309.0 • | 339.0 • | 368.0 • | 401.0 • | 424.0 • | 5.4 |
| Greedy$^+$ (Yaroslavtsev et al., 2020) | 309.0 • | 339.0 • | 368.0 • | 401.0 • | 425.0 • | 4.4 |
| 1-guess- Greedy$^+$ (Feldman et al., 2023) | 314.0 • | 352.0 • | 385.0 • | 412.0 • | 440.0 • | 2.6 |
| EVO-SMC (Zhu et al., 2024) | 292.6 ± 6.2 • | 325.8 ± 5.1 • | 355.3 ± 5.7 • | 381.9 ± 6.0 • | 410.5 ± 4.4 • | 8.0 |
| FPOMC (Bian et al., 2021) | 303.9 ± 4.3 • | 336.1 ± 3.9 • | 368.6 ± 4.4 • | 398.1 ± 3.9 • | 423.2 ± 5.4 • | 6.0 |
| EAMC (Bian et al., 2020) | 297.1 ± 3.2 • | 333.5 ± 5.0 • | 364.1 ± 5.1 • | 398.5 ± 4.1 • | 425.5 ± 5.7 • | 6.2 |
| POMC (Qian et al., 2017a) | 314.6 ± 1.8 • | 350.9 ± 2.4 • | 383.7 ± 2.6 • | 413.8 ± 2.6 • | 441.5 ± 2.4 • | 2.4 |
| EPOL (this paper) | **319.1 ± 0.8** | **356.6 ± 0.9** | **389.8 ± 0.6** | **419.0 ± 0.6** | **445.7 ± 0.6** | 1.0 |
| | *congress* (475 vertices, 13,289 edges) | | | | | |
| GGA (Zhang & Vorobeychik, 2016) | 326.0 • | 348.0 • | 368.0 • | 384.0 • | 398.0 • | 6.0 |
| Greedy$^+$ (Yaroslavtsev et al., 2020) | 326.0 • | 348.0 • | 368.0 • | 384.0 • | 399.0 • | 5.0 |
| 1-guess-Greedy$^+$ (Feldman et al., 2023) | 330.0 • | 355.0 • | 376.0 • | 394.0 • | 410.0 • | 2.2 |
| EVO-SMC (Zhu et al., 2024) | 309.3 ± 3.6 • | 327.8 ± 3.6 • | 348.3 ± 4.1 • | 361.9 ± 3.8 • | 376.5 ± 2.7 • | 7.0 |
| FPOMC (Bian et al., 2021) | 295.6 ± 9.3 • | 320.1 ± 7.6 • | 340.3 ± 9.7 • | 353.9 ± 10.3• | 374.1 ± 12.1• | 8.0 |
| EAMC (Bian et al., 2020) | 327.4 ± 2.5 • | 350.5 ± 2.0 • | 369.5 ± 1.4 • | 385.2 ± 1.9 • | 400.9 ± 2.0 • | 4.0 |
| POMC (Qian et al., 2017a) | 329.4 ± 1.1 • | 353.9 ± 1.1 • | 375.7 ± 2.1 • | 396.0 ± 3.2 • | 409.2 ± 3.3 • | 2.8 |
| EPOL (this paper) | **332.8 ± 0.4** | **358.0 ± 0.6** | **381.2 ± 0.6** | **399.9 ± 0.7** | **415.5 ± 0.5** | 1.0 |

*Table 7.* The objective value (number of covered vertices) of maximum coverage (avg $\pm$ std) obtained by the algorithms when $q = 10$ and the budgets $B \in \{300, 350, \ldots, 500\}$. For each $B$, the largest number is bolded, and '•/∘' denote that EPOL is significantly better/worse than the corresponding algorithm by the Wilcoxon signed-rank test with confidence level 0.05. Avg.R. denotes the average rank (the smaller, the better) of each algorithm under each setting.

| | *frb30-15-1* (450 vertices, 17,827 edges) | | | | | |
|---|---|---|---|---|---|---|
| Budget $B$ | 300 | 350 | 400 | 450 | 500 | Avg.R. |
| GGA (Zhang & Vorobeychik, 2016) | 331.0 • | 350.0 • | 371.0 • | 387.0 • | 406.0 • | 5.0 |
| Greedy$^+$ (Yaroslavtsev et al., 2020) | 331.0 • | 352.0 • | 372.0 • | 388.0 • | 406.0 • | 4.0 |
| 1-guess-Greedy$^+$ (Feldman et al., 2023) | 334.0 • | 357.0 • | 375.0 • | 392.0 • | 409.0 • | 3.0 |
| EVO-SMC (Zhu et al., 2024) | 301.1 ± 5.9 • | 323.4 ± 2.8 • | 346.2 ± 4.3 • | 364.7 ± 4.7 • | 381.6 ± 3.9 • | 8.0 |
| FPOMC (Bian et al., 2021) | 322.8 ± 6.5 • | 346.8 ± 3.9 • | 369.0 ± 2.3 • | 386.1 ± 3.2 • | 402.6 ± 3.5 • | 6.2 |
| EAMC (Bian et al., 2020) | 324.4 ± 2.5 • | 345.2 ± 3.5 • | 365.1 ± 3.3 • | 379.8 ± 2.0 • | 391.7 ± 4.8 • | 6.8 |
| POMC (Qian et al., 2017a) | 335.1 ± 1.2 • | 359.1 ± 2.0 • | 377.3 ± 1.9 • | 397.1 ± 3.2 • | 409.5 ± 2.6 • | 2.0 |
| EPOL (this paper) | **337.2 ± 1.0** | **362.0 ± 1.2** | **382.8 ± 0.7** | **401.3 ± 0.8** | **415.2 ± 0.9** | 1.0 |
| | *frb35-17-1* (595 vertices, 27,856 edges) | | | | | |
| GGA (Zhang & Vorobeychik, 2016) | 342.0 • | 377.0 • | 405.0 • | 433.0 • | 461.0 • | 5.0 |
| Greedy$^+$ (Yaroslavtsev et al., 2020) | 344.0 • | 377.0 • | 406.0 • | 433.0 • | 461.0 • | 4.0 |
| 1-guess-Greedy$^+$ (Feldman et al., 2023) | **359.0** | 392.0 • | 420.0 • | 446.0 • | 471.0 • | 2.4 |
| EVO-SMC (Zhu et al., 2024) | 321.2 ± 6.2 • | 351.2 ± 4.4 • | 373.4 ± 7.1 • | 404.4 ± 7.4 • | 428.4 ± 7.3 • | 8.0 |
| FPOMC (Bian et al., 2021) | 341.7 ± 2.5 • | 371.6 ± 5.7 • | 404.5 ± 3.9 • | 429.1 ± 5.1 • | 455.9 ± 3.8 • | 6.4 |
| EAMC (Bian et al., 2020) | 334.3 ± 2.6 • | 370.9 ± 4.1 • | 401.6 ± 4.3 • | 432.5 ± 4.2 • | 456.9 ± 8.2 • | 6.6 |
| POMC (Qian et al., 2017a) | 354.3 ± 2.7 • | 390.4 ± 2.8 • | 419.3 ± 2.0 • | 447.1 ± 2.3 • | 474.1 ± 2.6 • | 2.6 |
| EPOL (this paper) | **359.0 ± 0.0** | **395.9 ± 0.3** | **425.9 ± 0.7** | **454.0 ± 0.8** | **477.7 ± 0.6** | 1.0 |
| | *congress* (475 vertices, 13,289 edges) | | | | | |
| GGA (Zhang & Vorobeychik, 2016) | 420.0 • | 432.0 • | 442.0 • | 450.0 • | 457.0 • | 5.0 |
| Greedy$^+$ (Yaroslavtsev et al., 2020) | 420.0 • | 432.0 • | 442.0 • | 450.0 • | 457.0 • | 4.0 |
| 1-guess-Greedy$^+$ (Feldman et al., 2023) | 423.0 • | 437.0 • | 446.0 • | 453.0 • | 460.0 • | 2.6 |
| EVO-SMC (Zhu et al., 2024) | 386.0 ± 4.5 • | 398.9 ± 4.0 • | 409.3 ± 4.4 • | 415.8 ± 3.4 • | 421.5 ± 2.6 • | 7.0 |
| FPOMC (Bian et al., 2021) | 358.7 ± 10.5 • | 381.3 ± 6.1 • | 393.4 ± 9.0 • | 403.0 ± 11.3 • | 419.7 ± 6.0 • | 8.0 |
| EAMC (Bian et al., 2020) | 414.7 ± 1.1 • | 428.4 ± 2.0 • | 439.2 ± 1.7 • | 447.0 ± 2.1 • | 453.2 ± 1.0 • | 6.0 |
| POMC (Qian et al., 2017a) | 422.9 ± 0.8 • | 436.9 ± 1.5 • | 446.8 ± 1.7 • | 455.0 ± 1.0 • | 460.2 ± 0.9 • | 2.4 |
| EPOL (this paper) | **423.5 ± 0.5** | **437.1 ± 0.5** | **447.8 ± 0.7** | **455.7 ± 0.5** | **462.1 ± 0.7** | 1.0 |

*Table 8.* The objective value (influence spread) of influence maximization (avg $\pm$ std) obtained by the algorithms for the probability of each edge 0.05 and the budgets $B \in \{100, 200, \ldots, 500\}$. For each $B$, the largest number is bolded, and '•/∘' denote that EPOL is significantly better/worse than the corresponding algorithm by the Wilcoxon signed-rank test with confidence level 0.05. Avg.R. denotes the average rank (the smaller, the better) of each algorithm under each setting.

| Budget $B$ | 100 | 200 | 300 | 400 | 500 | Avg.R. |
|---|---|---|---|---|---|---|
| *graph100* (100 vertices, 3,465 edges) | | | | | | |
| GGA (Zhang & Vorobeychik, 2016) | 22.43 ± 1.21 • | 35.49 ± 1.35 • | 43.30 ± 0.99 • | 49.71 ± 1.10 • | 55.18 ± 0.89 • | 7.8 |
| Greedy⁺ (Yaroslavtsev et al., 2020) | 23.87 ± 0.37 • | 35.61 ± 0.78 • | 44.22 ± 0.63 • | 50.92 ± 0.61 • | 55.80 ± 0.69 • | 6.4 |
| 1-guess-Greedy⁺ (Feldman et al., 2023) | 24.66 ± 0.17 • | 37.17 ± 0.29 • | 45.70 ± 0.33 • | 51.89 ± 0.18 • | 56.91 ± 0.26 • | 4.0 |
| EVO-SMC (Zhu et al., 2024) | 25.03 ± 0.32 • | 37.39 ± 0.55 • | 44.72 ± 0.79 • | 50.45 ± 0.92 • | 54.44 ± 1.02 • | 5.6 |
| FPOMC (Bian et al., 2021) | 24.29 ± 0.64 • | 36.58 ± 1.12 • | 45.18 ± 0.95 • | 51.24 ± 0.70 • | 55.74 ± 0.88 • | 5.4 |
| EAMC (Bian et al., 2020) | 25.08 ± 0.27 • | 37.80 ± 0.18 | 45.88 ± 0.44 | 51.19 ± 0.56 • | 56.21 ± 0.34 • | 3.0 |
| POMC (Qian et al., 2017a) | 24.96 ± 0.35 • | 37.56 ± 0.39 • | 45.81 ± 0.30 • | 52.24 ± 0.21 • | 57.03 ± 0.22 • | 2.8 |
| EPOL (this paper) | **25.48 ± 0.17** | **37.98 ± 0.17** | **46.41 ± 0.24** | **52.76 ± 0.14** | **57.71 ± 0.17** | 1.0 |
| *graph200* (200 vertices, 9,950 edges) | | | | | | |
| GGA (Zhang & Vorobeychik, 2016) | 44.31 ± 1.14 • | 70.87 ± 2.84 • | 89.51 ± 2.55 • | 103.44 ± 2.64 • | 109.07 ± 3.16 • | 8.0 |
| Greedy⁺ (Yaroslavtsev et al., 2020) | 45.12 ± 0.47 • | 81.57 ± 0.72 • | 94.06 ± 0.91 • | 106.95 ± 1.01 • | 115.22 ± 1.88 • | 7.0 |
| 1-guess-Greedy⁺ (Feldman et al., 2023) | 46.11 ± 0.52 • | 82.56 ± 0.75 • | 96.58 ± 0.69 • | 108.97 ± 0.33 • | 117.47 ± 0.28 • | 5.4 |
| EVO-SMC (Zhu et al., 2024) | **48.04 ± 0.44** | 84.38 ± 0.21 • | 96.81 ± 0.60 • | 109.09 ± 1.66 • | 116.77 ± 1.78 • | 3.6 |
| FPOMC (Bian et al., 2021) | 46.72 ± 0.77 • | 82.62 ± 2.57 • | 95.92 ± 0.67 • | 108.54 ± 0.90 • | 117.64 ± 0.91 • | 5.2 |
| EAMC (Bian et al., 2020) | 47.92 ± 0.43 | **84.78 ± 0.44** | 97.06 ± 0.46 • | 110.81 ± 0.27 | 118.37 ± 2.06 • | 2.2 |
| POMC (Qian et al., 2017a) | 46.78 ± 0.61 • | 83.87 ± 0.66 • | 98.41 ± 0.59 • | 110.05 ± 0.73 • | 119.40 ± 0.24 • | 3.0 |
| EPOL (this paper) | 47.79 ± 0.33 | 84.76 ± 0.35 | **99.73 ± 0.26** | **111.04 ± 0.43** | **120.12 ± 0.26** | 1.6 |
| *insecta* (152 vertices, 6,716 edges) | | | | | | |
| GGA (Zhang & Vorobeychik, 2016) | 38.09 ± 1.33 • | 53.35 ± 4.24 • | 65.48 ± 3.31 • | 75.55 ± 1.39 • | 83.17 ± 1.49 • | 8.0 |
| Greedy⁺ (Yaroslavtsev et al., 2020) | 39.49 ± 1.07 • | 57.94 ± 0.52 • | 68.86 ± 2.65 • | 78.39 ± 1.31 • | 84.69 ± 1.04 • | 6.6 |
| 1-guess-Greedy⁺ (Feldman et al., 2023) | 40.57 ± 0.64 • | 58.99 ± 0.27 • | 72.90 ± 0.66 • | 80.24 ± 0.39 • | 86.59 ± 0.26 • | 3.6 |
| EVO-SMC (Zhu et al., 2024) | 41.76 ± 0.28 | 59.63 ± 0.29 | 68.73 ± 0.87 • | 78.14 ± 1.29 • | 85.11 ± 0.62 • | 5.0 |
| FPOMC (Bian et al., 2021) | 40.48 ± 0.57 • | 58.75 ± 0.39 • | 71.70 ± 1.82 • | 79.75 ± 0.41 • | 85.92 ± 0.89 • | 5.0 |
| EAMC (Bian et al., 2020) | **41.79 ± 0.18** | 59.66 ± 0.35 | 70.81 ± 1.18 • | 80.04 ± 0.68 • | 86.48 ± 0.41 • | 3.2 |
| POMC (Qian et al., 2017a) | 40.90 ± 0.51 • | 58.40 ± 0.57 • | 73.96 ± 0.40 • | 81.10 ± 0.42 • | 87.15 ± 0.28 • | 3.2 |
| EPOL (this paper) | 41.64 ± 0.26 | **59.89 ± 0.33** | **74.60 ± 0.20** | **81.77 ± 0.15** | **88.00 ± 0.07** | 1.4 |

*Table 9.* The objective value (influence spread) of influence maximization (avg $\pm$ std) obtained by the algorithms for the probability of each edge 0.1 and the budgets $B \in \{100, 200, \ldots, 500\}$. For each $B$, the largest number is bolded, and '•/∘' denote that EPOL is significantly better/worse than the corresponding algorithm by the Wilcoxon signed-rank test with confidence level 0.05. Avg.R. denotes the average rank (the smaller, the better) of each algorithm under each setting.

| Budget $B$ | 300 | 350 | 400 | 450 | 500 | Avg.R. |
|---|---|---|---|---|---|---|
| *graph100* (100 vertices, 3,465 edges) | | | | | | |
| GGA (Zhang & Vorobeychik, 2016) | 58.49 ± 0.93 • | 66.18 ± 1.78 • | 74.17 ± 0.87 • | 78.58 ± 1.03 • | 81.76 ± 0.88 • | 7.8 |
| Greedy⁺ (Yaroslavtsev et al., 2020) | 58.28 ± 0.38 • | 68.71 ± 0.97 • | 75.42 ± 0.63 • | 79.47 ± 0.58 • | 82.62 ± 0.42 • | 6.8 |
| 1-guess-Greedy⁺ (Feldman et al., 2023) | 59.66 ± 0.26 • | 71.49 ± 0.25 • | 76.79 ± 0.22 • | 80.33 ± 0.17 • | 83.36 ± 0.19 • | 3.6 |
| EVO-SMC (Zhu et al., 2024) | 60.22 ± 0.41 • | 70.34 ± 1.26 • | 75.29 ± 1.01 • | 79.81 ± 0.28 • | 82.53 ± 0.56 • | 5.8 |
| FPOMC (Bian et al., 2021) | 59.05 ± 0.90 • | 70.44 ± 1.16 • | 76.40 ± 0.43 • | 80.20 ± 0.62 • | 83.07 ± 0.43 • | 4.8 |
| EAMC (Bian et al., 2020) | 60.40 ± 0.35 | 71.02 ± 0.74 • | 75.93 ± 0.90 • | 80.07 ± 0.41 • | 83.79 ± 0.37 • | 3.8 |
| POMC (Qian et al., 2017a) | 59.97 ± 0.16 • | 71.77 ± 0.16 • | 76.92 ± 0.12 • | 80.83 ± 0.27 • | 83.80 ± 0.16 • | 2.4 |
| EPOL (this paper) | **60.65 ± 0.18** | **72.13 ± 0.12** | **77.28 ± 0.06** | **81.37 ± 0.09** | **84.09 ± 0.05** | 1.0 |
| *graph200* (200 vertices, 9,950 edges) | | | | | | |
| GGA (Zhang & Vorobeychik, 2016) | 115.79 ± 0.93 • | 148.37 ± 3.00 • | 159.45 ± 1.22 • | 166.47 ± 2.82 • | 168.26 ± 1.58 • | 8.0 |
| Greedy⁺ (Yaroslavtsev et al., 2020) | 116.14 ± 1.35 • | 153.05 ± 0.76 • | 162.60 ± 0.64 • | 169.17 ± 0.88 • | 172.00 ± 1.10 • | 7.0 |
| 1-guess-Greedy⁺ (Feldman et al., 2023) | 118.53 ± 0.47 • | 154.32 ± 0.57 • | 163.57 ± 0.17 • | 170.43 ± 0.21 • | 173.87 ± 0.45 • | 5.4 |
| EVO-SMC (Zhu et al., 2024) | 120.36 ± 0.30 | 155.23 ± 0.12 • | 163.78 ± 0.22 • | 169.63 ± 0.89 • | 172.37 ± 1.35 • | 4.6 |
| FPOMC (Bian et al., 2021) | 119.88 ± 0.35 • | 155.26 ± 0.17 • | 163.82 ± 0.26 • | 170.70 ± 0.35 • | 174.39 ± 0.57 • | 3.4 |
| EAMC (Bian et al., 2020) | **120.66 ± 0.45** | 155.32 ± 0.18 | 163.93 ± 0.22 | 170.79 ± 0.45 • | 174.36 ± 0.75 • | 2.2 |
| POMC (Qian et al., 2017a) | 119.65 ± 0.25 • | 154.91 ± 0.20 • | 163.54 ± 0.18 • | 170.70 ± 0.35 • | 174.72 ± 0.07 • | 4.2 |
| EPOL (this paper) | 120.61 ± 0.19 | **155.55 ± 0.28** | **164.15 ± 0.29** | **171.14 ± 0.09** | **174.99 ± 0.09** | 1.2 |
| *insecta* (152 vertices, 6,716 edges) | | | | | | |
| GGA (Zhang & Vorobeychik, 2016) | 95.34 ± 1.07 • | 107.39 ± 2.65 • | 115.54 ± 1.53 • | 122.04 ± 1.04 • | 126.28 ± 1.00 • | 8.0 |
| Greedy⁺ (Yaroslavtsev et al., 2020) | 95.85 ± 0.83 • | 111.41 ± 0.96 • | 120.15 ± 1.75 • | 124.18 ± 0.44 • | 128.16 ± 0.40 • | 6.6 |
| 1-guess-Greedy⁺ (Feldman et al., 2023) | 97.94 ± 0.34 • | 113.09 ± 0.38 • | 121.65 ± 0.25 • | 125.13 ± 0.31 • | 129.05 ± 0.18 • | 4.8 |
| EVO-SMC (Zhu et al., 2024) | 99.37 ± 0.36 • | 113.54 ± 0.56 • | 119.45 ± 0.74 • | 124.55 ± 1.08 • | 128.02 ± 0.76 • | 5.2 |
| FPOMC (Bian et al., 2021) | 98.83 ± 0.29 • | 112.95 ± 0.52 • | 122.11 ± 0.13 • | 125.50 ± 0.47 • | 129.33 ± 0.27 • | 4.0 |
| EAMC (Bian et al., 2020) | 99.28 ± 0.31 • | 113.93 ± 0.18 • | 120.58 ± 1.01 • | 126.21 ± 0.53 • | 128.80 ± 0.42 • | 3.6 |
| POMC (Qian et al., 2017a) | 98.84 ± 0.31 • | 113.86 ± 0.33 • | 122.06 ± 0.21 • | 126.43 ± 0.20 • | 129.52 ± 0.13 • | 2.8 |
| EPOL (this paper) | **99.52 ± 0.18** | **114.25 ± 0.15** | **122.38 ± 0.11** | **126.69 ± 0.08** | **129.71 ± 0.09** | 1.0 |

## B.3. Results of Sto-EVO-SMC

Sto-EVO-SMC is a stochastic version of EVO-SMC, which brings two parameters $\epsilon$ and $p$, maintaining the same guarantee as EVO-SMC with probability $1 - \epsilon$. We run sto-EVO-SMC by setting $\epsilon \in \{0.1, 0.2, 0.5\}$ and $p \in \{0.2, 0.5\}$, and plot the average results of sto-EVO-SMC-$\epsilon$-$p$, EVO-SMC and EPOL in Figure 1. We can find that all the class of EVO-SMC algorithms (i.e., EVO-SMC and sto-EVO-SMC-$\epsilon$-$p$) shows minimal difference across parameter settings, and performs worse than EPOL in all cases, expect for $B = 100$ on *graph200* and *insecta*.

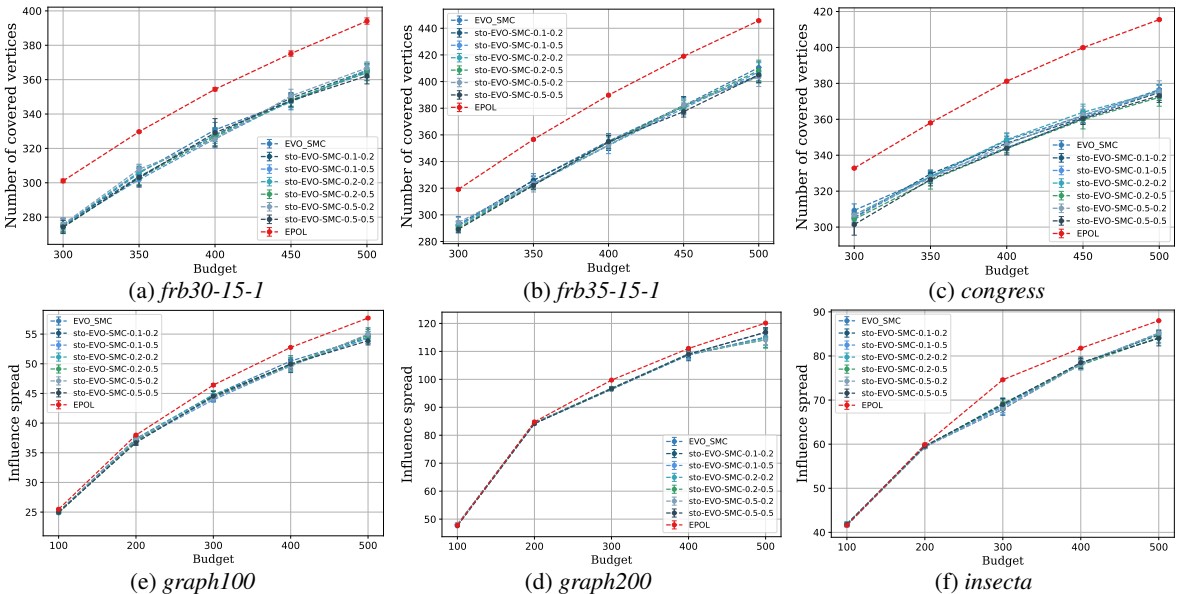

*Figure 1.* The average objective values of sto-EVO-SMC-$\epsilon$-$p$, EVO-SMC, and EPOL: number of covered vertices (top) for maximum coverage with $q = 5$ and influence spread (bottom) for influence maximization with the probability of each edge 0.05, where $\epsilon \in \{0.1, 0.2, 0.5\}$ and $p \in \{0.2, 0.5\}$.

## B.4. Results of P-POMC

*Table 10.* The objective value (avg $\pm$ std) obtained by EPOL and P-POMC for $q = 5$ on maximum coverage and the probability of each edge 0.05 on influence maximization. For each $B$, the largest number is bolded, and '•/○' denote that EPOL is significantly better/worse than the corresponding algorithm by the Wilcoxon signed-rank test with confidence level 0.05.

| | | *frb30-15-1* (450 vertices, 17,827 edges) | | | | |
|---|---|---|---|---|---|---|
| | $B$ | 300 | 350 | 400 | 450 | 500 |
| | P-POMC | $299.6 \pm 1.0$ • | $329.5 \pm 0.8$ | $353.9 \pm 1.3$ | $375.0 \pm 1.9$ | $393.5 \pm 1.5$ |
| | EPOL | $\mathbf{301.1 \pm 1.0}$ | $\mathbf{329.7 \pm 0.5}$ | $\mathbf{354.4 \pm 0.9}$ | $\mathbf{375.2 \pm 1.5}$ | $\mathbf{394.1 \pm 1.8}$ |
| **Maximum coverage** | | *frb35-17-1* (595 vertices, 27,856 edges) | | | | |
| | P-POMC | $318.1 \pm 0.3$ • | $355.0 \pm 0.4$ • | $389.5 \pm 1.0$ | $418.2 \pm 0.7$ • | $445.6 \pm 0.7$ |
| | EPOL | $\mathbf{319.1 \pm 0.8}$ | $\mathbf{356.6 \pm 0.9}$ | $\mathbf{389.8 \pm 0.6}$ | $\mathbf{419.0 \pm 0.6}$ | $\mathbf{445.7 \pm 0.6}$ |
| | | *congress* (475 vertices, 13,289 edges) | | | | |
| | P-POMC | $330.3 \pm 0.8$ • | $355.6 \pm 0.5$ • | $380.0 \pm 1.3$ • | $399.0 \pm 1.0$ | $414.6 \pm 0.8$ • |
| | EPOL | $\mathbf{332.8 \pm 0.4}$ | $\mathbf{358.0 \pm 0.6}$ | $\mathbf{381.2 \pm 0.6}$ | $\mathbf{399.9 \pm 0.7}$ | $\mathbf{415.5 \pm 0.5}$ |
| | | *graph100* (100 vertices, 3,465 edges) | | | | |
| | $B$ | 100 | 200 | 300 | 400 | 500 |
| | P-POMC | $25.07 \pm 0.14$ • | $37.97 \pm 0.22$ | $46.29 \pm 0.13$ | $52.75 \pm 0.19$ | $57.47 \pm 0.11$ • |
| | EPOL | $\mathbf{25.48 \pm 0.17}$ | $\mathbf{37.98 \pm 0.17}$ | $\mathbf{46.41 \pm 0.24}$ | $\mathbf{52.76 \pm 0.14}$ | $\mathbf{57.71 \pm 0.17}$ |
| **Influence maximization** | | *graph200* (200 vertices, 9,950 edges) | | | | |
| | P-POMC | $47.33 \pm 0.34$ • | $84.39 \pm 0.32$ • | $99.53 \pm 0.21$ | $110.98 \pm 0.17$ | $120.02 \pm 0.29$ |
| | EPOL | $\mathbf{47.79 \pm 0.33}$ | $\mathbf{84.76 \pm 0.35}$ | $\mathbf{99.73 \pm 0.26}$ | $\mathbf{111.04 \pm 0.43}$ | $\mathbf{120.12 \pm 0.26}$ |
| | | *insecta* (152 vertices, 6,716 edges) | | | | |
| | P-POMC | $41.59 \pm 0.31$ | $59.67 \pm 0.39$ | $74.47 \pm 0.24$ | $81.75 \pm 0.17$ | $87.94 \pm 0.24$ |
| | EPOL | $\mathbf{41.64 \pm 0.26}$ | $\mathbf{59.89 \pm 0.33}$ | $\mathbf{74.60 \pm 0.20}$ | $\mathbf{81.77 \pm 0.15}$ | $\mathbf{88.00 \pm 0.07}$ |

To evaluate the effectiveness of EPOL, we compare it with a variant called P-POMC. In P-POMC, the original problem runs on $K_B$ parallel processors instead of solving $K_B$ independent residual problems. The best feasible solution among

these $K_B$ processors is used as the final output of a single P-POMC run. By running P-POMC 10 times and comparing the average results with EPOL, we observe that EPOL consistently outperforms P-POMC. Moreover, EPOL demonstrates a significant advantage over P-POMC in certain cases, as marked in Table 10. This comparison underscores the effectiveness of EPOL across various scenarios.

### B.5. Results of EPOL-full

Although the setting that EPOL runs a subset of residual problems ($K_B$ residual problems) is sufficient to show the superiority of the proposed EPOL as the implemented version is weaker, we conduct additional experiments comparing EPOL (as in the previous experiments) and the full version (EPOL-full), which enumerates all residual problems. For $q = 5$, the objective values (avg $\pm$ std) on maximum coverage for three datasets are summarized in Table 11. The results show that EPOL-full consistently outperforms EPOL with significant advantages in several cases, highlighting EPOL-full's potential to improve performance by addressing all residual problems.

*Table 11.* The objective value (number of covered vertices) of maximum coverage (avg $\pm$ std) obtained by EPOL and EPOL-full when $q = 5$ and the budgets $B \in \{300, 350, \ldots, 500\}$. For each $B$, the larger number is bolded, and '$\bullet/\circ$' denote that EPOL-full is significantly outperforms EPOL by the Wilcoxon signed-rank test with confidence level 0.05.

| | | | | | |
|---|---|---|---|---|---|
| *frb-30-15-1* (450 vertices, 17,827 edges) | | | | | |
| Budget $B$ | 300 | 350 | 400 | 450 | 500 |
| EPOL | $301.1 \pm 1.0$ $\bullet$ | $329.7 \pm 0.5$ $\bullet$ | $354.4 \pm 0.9$ | $375.2 \pm 1.5$ $\bullet$ | $394.1 \pm 1.8$ |
| EPOL-full | $\mathbf{302.1 \pm 0.8}$ | $\mathbf{330.9 \pm 0.3}$ | $\mathbf{354.8 \pm 0.4}$ | $\mathbf{377.3 \pm 1.3}$ | $\mathbf{395.2 \pm 1.1}$ |
| *frb-35-17-1* (595 vertices, 27,856 edges) | | | | | |
| EPOL | $319.1 \pm 0.8$ | $356.6 \pm 0.9$ $\bullet$ | $389.8 \pm 0.6$ $\bullet$ | $\mathbf{419.0 \pm 0.6}$ | $445.7 \pm 0.6$ $\bullet$ |
| EPOL-full | $\mathbf{319.8 \pm 0.4}$ | $\mathbf{357.9 \pm 0.3}$ | $\mathbf{390.6 \pm 0.5}$ | $419.0 \pm 0.0$ | $\mathbf{446.5 \pm 0.5}$ |
| *congress* (475 vertices, 13,289 edges) | | | | | |
| EPOL | $332.8 \pm 0.4$ | $358.0 \pm 0.6$ $\bullet$ | $381.2 \pm 0.6$ | $399.9 \pm 0.7$ | $415.5 \pm 0.5$ $\bullet$ |
| EPOL-full | $\mathbf{333.4 \pm 0.5}$ | $\mathbf{359.0 \pm 0.6}$ | $\mathbf{381.4 \pm 0.7}$ | $\mathbf{400.2 \pm 1.0}$ | $\mathbf{416.1 \pm 0.5}$ |

### B.6. Objective Values vs. Runtime

The class of greedy algorithms, including GGA, Greedy$^+$, and 1-guess Greedy$^+$, is composed of fixed-time algorithms with a runtime complexity of $O(nK_B)$, $O(nK_B)$ and $O(n^2 K_B)$, respectively. In contrast, other algorithms are anytime algorithms, whose performance improves with increased runtime. EPOL surpasses all greedy algorithms within $12nK_B$ for maximum coverage and $4nK_B$ for influence maximization, as shown in Figures 2 and 3. Moreover, EPOL eventually converges to the best objective value in almost all cases.

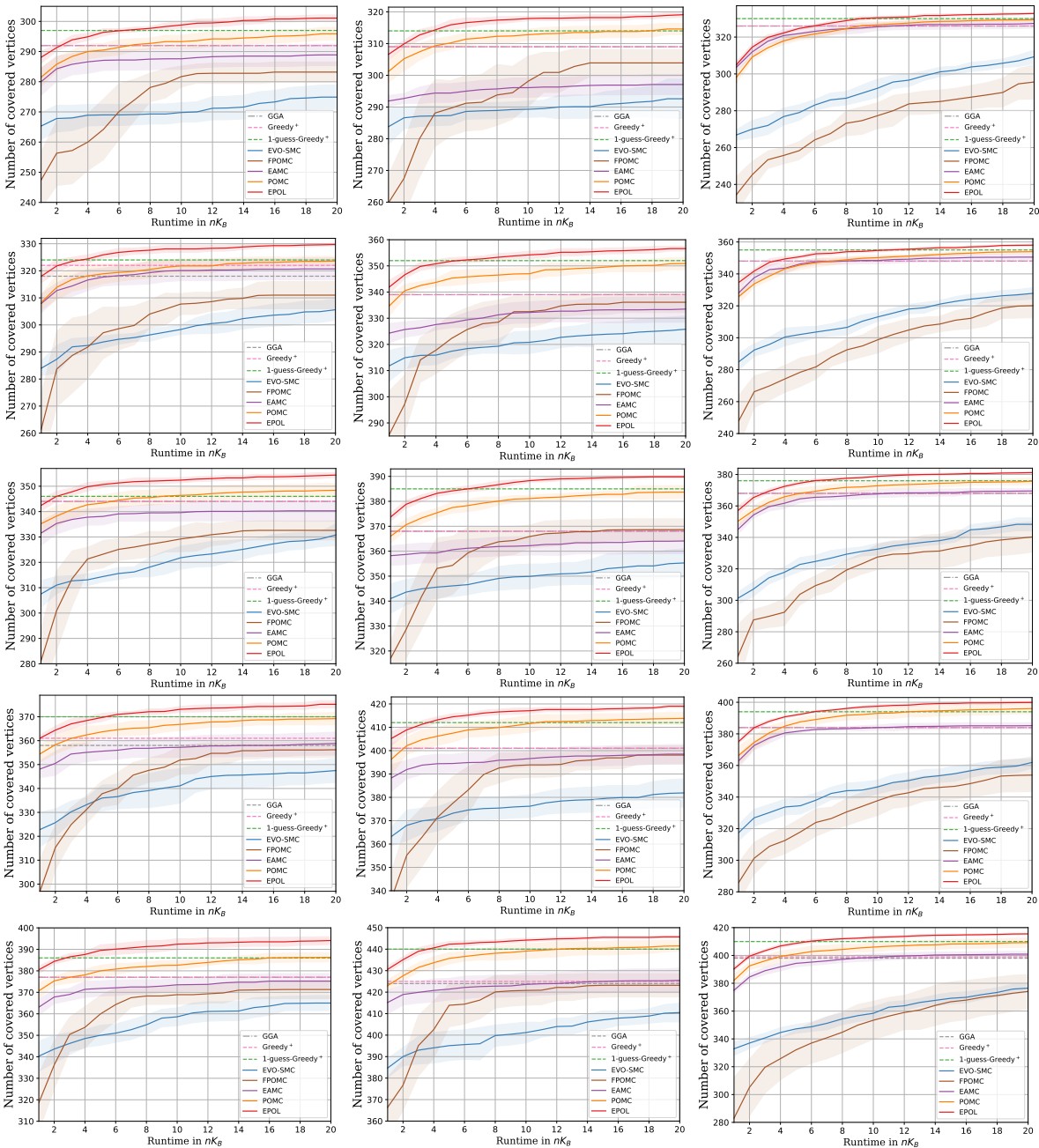

*Figure 2.* The average objective value (number of covered vertices) vs. runtime (i.e., number of objective evaluations) for maximum coverage with $q = 5$ and budget $B \in \{300, 350, 400, 450, 500\}$ (from top to bottom) on datasets *frb-30-15-1* (left), *frb35-17-1* (middle), and *congress* (right). Error bars show standard deviations.

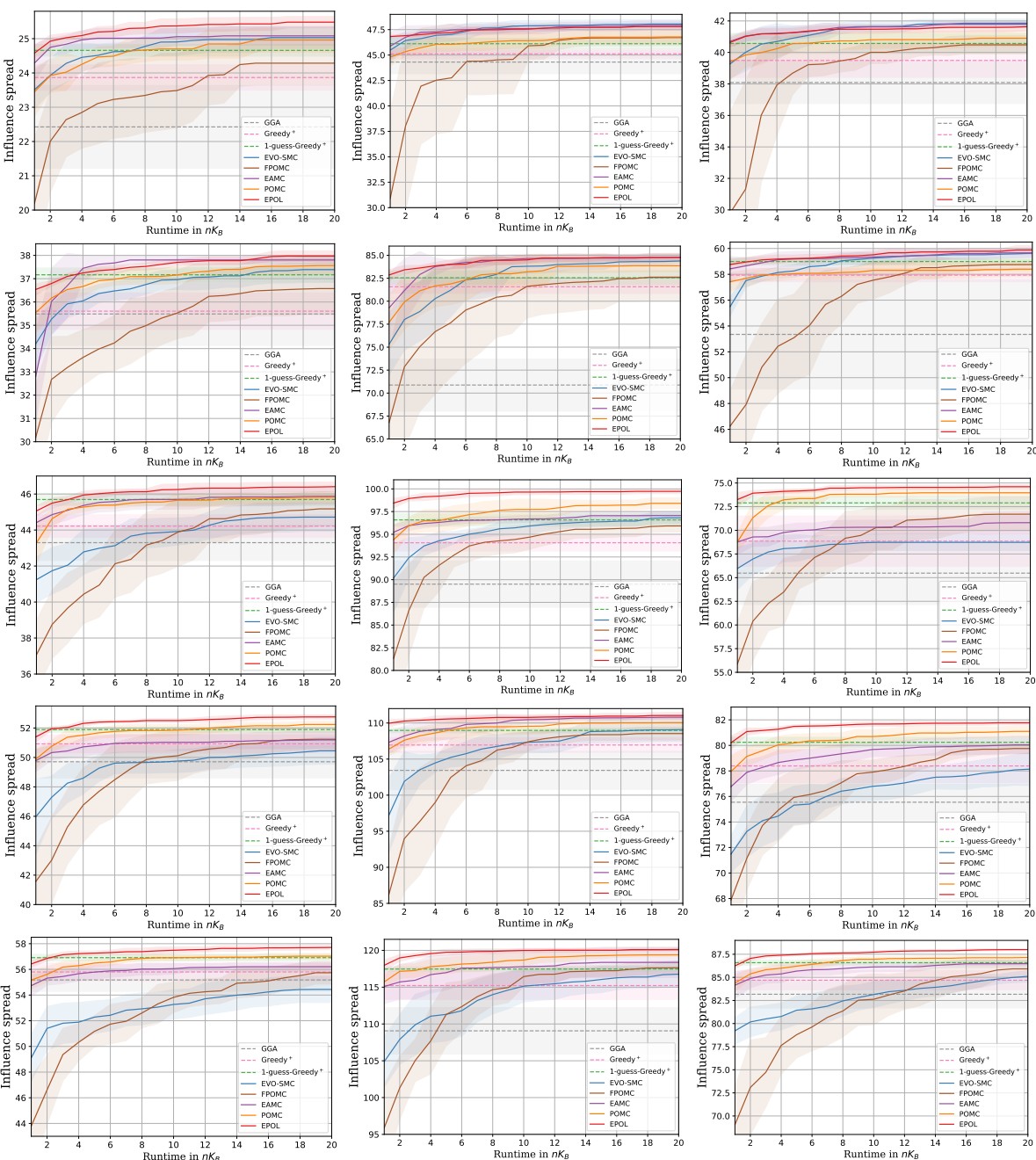

*Figure 3.* The average objective value (influence spread) vs. runtime (i.e., number of objective evaluations) for influence maximization with the probability of each edge 0.05 and budget $B \in \{100, 200, 300, 400, 500\}$ (from top to bottom) on datasets *graph100* (left), *graph200* (middle), and *insecta* (right). Error bars show standard deviations.

