# OpenReview forum: "Improved Theoretically-Grounded Evolutionary Algorithms for Subset Selection with a Linear Cost Constraint"
_ICML.cc/2025/Conference — ICML 2025 poster_

### Official Review · Reviewer_sMVc · 2025-02-24

**Overall Recommendation:** 4

**Summary:**

The manuscript presents an advanced study on subset selection problems, which are prevalent in various fields such as machine learning, operations research, and economics. The authors focus on subset selection under a linear cost constraint, a problem characterized by its NP-hardness and practical importance. The paper introduces an improved theoretically-grounded evolutionary algorithm (EA), enhancing the performance of the Pareto Optimized Multi-Criteria (POMC) algorithm and proposing a novel multi-objective EA, named EPOL.

**Claims And Evidence:**

yes, this paper has made complete proofs for all Lemmas.

**Essential References Not Discussed:**

this paper provides a theoretical understanding of EAs, which has not been discussed in previous literatures.

**Experimental Designs Or Analyses:**

yes, the experimental designs and analyses are abundant.

**Methods And Evaluation Criteria:**

yes, this work  deepens the theoretical understanding of EAs.

**Other Comments Or Suggestions:**

None

**Other Strengths And Weaknesses:**

Strengths:

The paper makes a significant theoretical contribution by improving the approximation guarantee of the POMC algorithm to 1/2, which is a substantial advancement in the field.

The proposed EPOL algorithm is innovative and achieves the best-known practical approximation guarantee of 0.6174, which is a notable empirical performance.

This work not only advances the subset selection problem but also deepens the theoretical understanding of EAs.

The writing is clear and the mathematical formulation is precise, making the paper accessible to readers familiar with evolutionary algorithms and optimization.

Weaknesses:

The manuscript would benefit from a more detailed comparison with other state-of-the-art algorithms in the literature, especially in terms of computational complexity and scalability.

The discussion on the limitations of the proposed methods is somewhat brief. A more in-depth analysis could provide insights into potential improvements or alternative applications.

I understand that this work is mainly a theoretical contribution, but if we can compare more SOTA algorithms, we can further demonstrate the superiority of the algorithms.

**Questions For Authors:**

How does the proposed EPOL algorithm scale with the size of the problem (e.g., larger ground sets or higher-dimensional feature spaces)? Could the authors provide some insights or preliminary results on scalability?


Are there any parameters in the EPOL algorithm that require fine-tuning? If so, how sensitive are the results to these parameters?

The empirical study focuses on maximum coverage and influence maximization. Would the authors consider extending the study to other relevant problems such as budgeted submodular optimization?

**Relation To Broader Scientific Literature:**

yes, there remains a gap in the approximation bounds of EAs compared to greedy algorithms, and their full theoretical potential is yet to be realized.

**Theoretical Claims:**

yes, i have checked the proofs of Lemma 3.2 and 3.3

---

> ### Author Rebuttal · Authors · 2025-04-01
>
> Thank you very much for your positive feedback and for recognizing the strengths of our work. We appreciate your kind words regarding our contributions and writing. Please find our detailed responses below.
>
> ---
>
> > I understand that this work is mainly a theoretical contribution, but if we can compare more SOTA algorithms, we can further demonstrate the superiority of the algorithms.
>
> Thank you for your valuable suggestion. To the best of our knowledge, we have already included all state-of-the-art algorithms relevant to our problem setting in our comparisons. Nonetheless, we appreciate your perspective and will explore further comparisons in future work as more methods become available. Thank you.
>
> ---
>
> > How does the proposed EPOL algorithm scale with the size of the problem (e.g., larger ground sets or higher-dimensional feature spaces)? Could the authors provide some insights or preliminary results on scalability?
>
> Thank you for your insightful comment. The proposed algorithm is designed to be highly parallelizable, which makes it well-suited for larger-scale problems. Thanks to your suggestion, we compare EPOL against three comparison algorithms—namely, the baseline GGA, the high-performance greedy algorithm 1-guess-Greedy$^+$, and POMC—using a larger network dataset email-Eu-core (1005 vertices, 25,571 edges) on the maximum coverage problem with $q = 5$. For each budget $B$, we calculated the objective values (average $\pm$ std) and highlighted the best performance in bold. Additionally, a ‘$\bullet$’ indicates that EPOL significantly outperforms the corresponding algorithm, as confirmed by a Wilcoxon signed-rank test (confidence level 0.05). Consistent with the results reported in the paper, EPOL significantly outperforms other algorithms. Thank you again for your valuable feedback.
>
> |     Budget $B$     |           $300$           |           $350$           |
> | :----------------: | :-----------------------: | :-----------------------: |
> |        GGA         |       525 $\bullet$       |       575 $\bullet$       |
> | 1-guess-Greedy$^+$ |       528 $\bullet$       |       578 $\bullet$       |
> |        POMC        | 531.3 $\pm$ 1.3 $\bullet$ | 581.3 $\pm$ 0.9 $\bullet$ |
> |        EPOL        |    **532.8 $\pm$ 0.4**    |    **582.5 $\pm$ 0.5**    |
>
> ---
>
> > Are there any parameters in the EPOL algorithm that require fine-tuning? If so, how sensitive are the results to these parameters?
>
> We would like to clarify that the design of the EPOL algorithm inherently avoids the need for fine-tuning of parameters. This feature contributes to its robust and consistent performance across various settings. Thank you.
>
> ---
>
> > The empirical study focuses on maximum coverage and influence maximization. Would the authors consider extending the study to other relevant problems such as budgeted submodular optimization?
>
> Thank you for your insightful comment. While our empirical study has primarily focused on maximum coverage and influence maximization, our methodology is inherently adaptable to budgeted submodular optimization problems in scenarios where cost-aware resource allocation and the diminishing returns property of objective functions are central concerns. For instance: sensor placement, which balances information gain with installation costs (Krause et al., 2006); recommendation systems, which promote products within advertising budgets and respect user preferences (Ashkan et al., 2015); data summarization, which maximizes information retention under computational resource constraints (Lin & Bilmes, 2011); active learning, which selects maximally informative data samples under limited annotation budgets (Golovin & Krause, 2011); and human-assisted learning, which optimizes machine learning models with limited expert resources (De et al., 2020, 2021). As suggested by Reviewer vZX8, we will revise the paper to include more discussions on the applications of the studied problem, thereby improving its visibility. Thank you very much.
>
> Reference:
>
> Krause, A., et al. "Near-optimal sensor placements: Maximizing information while minimizing communication cost." Proc. IPSN, 2006.
>
> Ashkan, A., et al. "Optimal Greedy Diversity for Recommendation." Proc. IJCAI, 2015.
>
> Lin, H., and Bilmes, J. "A class of submodular functions for document summarization." Proc. ACL, 2011.
>
> Golovin, D., and Krause, A. "Adaptive submodularity: Theory and applications in active learning and stochastic optimization." J. AI Research, 2011.
>
> De, A., et al. "Regression under human assistance." Proc. AAAI, 2020.
>
> De, A., et al. "Classification under human assistance." Proc. AAAI, 2021.

---

### Official Review · Reviewer_vZX8 · 2025-03-10

**Overall Recommendation:** 4

**Summary:**

This paper contributed to 1) analyzing the existing approach for submodular optimization with a linear inequality constraint, and 2) proposing a novel approach with a better approximation guarantee and better practical performance. This paper first analyzed the existing approach, POMC, an evolutionary algorithm that maintains a population of non-dominated solutions where the constraint is treated as the second objective. The analysis of this paper improves the existing theoretical guarantee by a constant factor (from 0.5(1 - 1/e) to 0.5). The rigorous proof is provided in the main text, followed by a description of the difference in the proof from the proof of the existing result. Then, the authors propose a novel approach, EPOL, which simply repeats running POMC for subproblems that are constructed by removing each element. The theoretical analysis revealed the improved theoretical guarantee (from 0.5 to 0.61...). The empirical study shows that the proposed approach is not only theoretically improved, but also performs better than existing approaches over different test problems.

## update after rebuttal

All my concerns have been addressed during the rebuttal phase. I keep my initial score.

**Claims And Evidence:**

Their claims, improving theoretical guarantee of EAs for subset selection problem with a monotone and submodular objective function under a linear cost constraint as well as improving their practical performance, are supported by rigorous theoretical results and thorough experimental results.

**Essential References Not Discussed:**

No

**Experimental Designs Or Analyses:**

Empirical results only show the final performance. It seems that different algorithms spent different iterations / function calls / time. Ideally, to compare the performance, the budget (iterations/f-calls/time, etc.) should be the same for all algorithms. Low performance algorithms with fast convergence may be trivially improved by incorporating a restart mechanism.

As I am not familiar with this topic itself, I am not sure whether the experiments done in this paper meet the standard of the community.

**Methods And Evaluation Criteria:**

Methodologies and Experimentation looks valid. However, honestly, as I am not familiar with this specific problem class, I am not fully sure about my assessment.

**Other Comments Or Suggestions:**

The sizes of parentheses could be reconsidered for better presentation.

**Other Strengths And Weaknesses:**

Strengths:

- Improved theoretical results with rigorous and non-trivial proofs

- Strong empirical results showing superior performance of the proposed approach over different baselines on different problems

Weaknesses:

- The theoretical guarantee doesn't seem to match the practical performance, hence the improvement in theoretical guarantee does not necessarily imply the improvement in practice, hence its value is a bit questionable.

- Empirical results only show the final performance. It seems that different algorithms spent different iterations / function calls / time. Ideally, to compare the performance, the budget (iterations/f-calls/time, etc.) should be the same for all algorithms. Low performance algorithms with fast convergence may be trivially improved by incorporating a restart mechanism.

**Questions For Authors:**

Please read the “Other Strengths And Weaknesses” part.

**Relation To Broader Scientific Literature:**

The discussion about the potential real-world applications would improve the visibility of this paper.

**Theoretical Claims:**

I did check the proof and didn’t find any flaw.

---

> ### Author Rebuttal · Authors · 2025-04-01
>
> We sincerely appreciate your positive feedback and encouraging validation of our work's strengths. Please find our detailed responses below.
>
> ---
>
> > The discussion about the potential real-world applications would improve the visibility of this paper.
>
> The discussion about the potential real-world applications would enhance the impact and relevance of this paper. The problem studied in this paper is highly relevant to domains where cost-aware resource allocation and the diminishing returns property of objective functions play a key role. It has applications in diverse areas, e.g., combinatorial optimization, computer networks, data mining, and machine learning. Beyond the maximum coverage and influence maximization examined in this paper, potential applications include sensor placement, balancing information gain with installation costs (Krause et al., 2006); recommendation systems, promoting products within advertising budgets while respecting user preferences (Ashkan et al., 2015); data summarization, maximizing information retention under computational resource constraints (Lin & Bilmes, 2011); active learning, selecting maximally informative data samples under limited annotation budgets (Golovin & Krause, 2011); and human-assisted learning, optimizing machine learning models with limited expert resources (De et al., 2020, 2021). We appreciate your suggestion and will incorporate these discussions into the paper. Thank you very much for your suggestion.
>
> Reference:
>
> Krause, A., et al. "Near-optimal sensor placements: Maximizing information while minimizing communication cost." Proc. IPSN, 2006.
>
> Ashkan, A., et al. "Optimal Greedy Diversity for Recommendation." Proc. IJCAI, 2015.
>
> Lin, H., and Bilmes, J. "A class of submodular functions for document summarization." Proc. ACL, 2011.
>
> Golovin, D., and Krause, A. "Adaptive submodularity: Theory and applications in active learning and stochastic optimization." J. AI Research, 2011.
>
> De, A., et al. "Regression under human assistance." Proc. AAAI, 2020.
>
> De, A., et al. "Classification under human assistance." Proc. AAAI, 2021.
>
> ---
>
> > Empirical results only show the final performance. It seems that different algorithms spent different iterations /function calls / time. Ideally, to compare the performance, the budget (iterations/f-calls/time, etc.) should be the same for all algorithms. Low performance algorithms with fast convergence may be trivially improved by incorporating a restart mechanism.
>
> Thank you for your comment. We fully agree, and actually have tried to make the comparison fair. The algorithms in our study fall into two categories: fixed-time algorithms such as GGA, Greedy$^+$, and 1-guess Greedy$^+$, with runtime complexities of $O(nK_B)$, $O(nK_B)$, and $O(n^2K_B)$, respectively, and anytime algorithms such as POMC, EAMC, FPOMC, and EVO-SMC, whose performance improves with increased runtime. To ensure a fair comparison, we allocated the same number of objective function evaluations ($20nK_B$) to all anytime algorithms. EPOL decomposes the original problem into $K_B$ subproblems and solves them in parallel using POMC, allocating a budget of $20nK_B$ evaluations to each subproblem. Furthermore, we also compared EPOL with a variant of POMC, called P-POMC, as described in Appendix B.4. P-POMC directly runs the original problem on $K_B$ parallel processors, where each processor is also allocated a budget of $20nK_B$ evaluations. Our experimental settings align with prior work in the literature [Bian et al., 2020, Roostapour et al., 2022, Zhu et al., 2024]. It is worth noting that fixed-time algorithms (GGA, Greedy$^+$, and 1-guess Greedy$^+$) cannot benefit from restart mechanisms, as they always return the same solution for a given input. We will revise to make them clear. Thank you very much.
>
> ---
>
> > The theoretical guarantee doesn't seem to match the practical performance, hence the improvement in theoretical guarantee does not necessarily imply the improvement in practice, hence its value is a bit questionable.
>
> Good point! Yes, the theoretical guarantee of an algorithm may not match its practical performance. This is because the theoretical guarantee implies the approximation performance in the worst case, which may differ from the tested real-world cases. Thus, it is possible that two algorithms with the same theoretical guarantee may demonstrate different performances in practice. However, a good theoretical guarantee is still important, which guarantees the worst-case performance of an algorithm and can characterize the robustness of the algorithm. Thus, a good algorithm should have both good theoretical guarantee and empirical performance. Our proposed algorithm EPOL has this property, which not only achieves the best-known practical theoretical guarantee, but also demonstrates empirical advantages across different settings. We will revise to add more explanation to make them clear. Thank you very much for your suggestion.

---

### Official Review · Reviewer_JmGW · 2025-03-11

**Overall Recommendation:** 4

**Summary:**

This paper studies the problem of subset selection with a linear cost constraint. The authors first improved the approximation guarantee (from (1-1/e)/2 to 1/2) of an existing evolutionary algorithm, so-called POMC, with the best empirical performance. Then, they proposed a new evolutionary algorithm EPOL with a better approximation guarantee of 0.6174. They also performed experiments on two applications (maximum coverage and influence maximization), showing that EPOL can achieve significantly better performance in most cases.

**Claims And Evidence:**

Yes, the claims made in the submission are supported by clear and convincing evidence. The improved approximation guarantee of the existing algorithm POMC and the approximation guarantee of the newly proposed algorithm EPOL are proved mathematically and correctly. The best empirical performance of EPOL has been clearly shown by extensive experiments. The authors compared EPOL with a large number of previous methods on the studied subset selection problem.

**Essential References Not Discussed:**

The related works are properly cited.

**Experimental Designs Or Analyses:**

Yes, I checked the experimental parts. The proposed algorithm is compared with a series of existing algorithms for the studied subset selection problem, including greedy and evolutionary algorithms. Two applications of subset selection with various parameter settings are tested. The authors performed a significance test, showing that the proposed EPOL algorithm is significantly better in most cases, and never significantly worse. They also conducted some ablation studies, e.g., comparing with multiple runs of the POMC algorithm on the original problem.

**Methods And Evaluation Criteria:**

The studied problem of subset selection with a linear cost constraint, where the objective function is monotone submodular, has wide applications and has been studied extensively. This work improves the approximation guarantee of a previous algorithm (with the best observed performance in experiments) and proposes a new better algorithm both theoretically and empirically. The theoretical analysis is rigorous, and the experiments use common benchmark data sets and compare all state-of-the-art works.  I think the proposed methods and/or evaluation criteria make sense.

**Other Comments Or Suggestions:**

The paper is well written. Some minor issues:

- page 1, second column, line 38-40, “further optimizes this approach” -> “further improves this approach”

- page 2, first column, line 107, “objective” -> “objective evaluation”

- page 4, second column, line 183, “Lemma 3.2” -> “The proof of Lemma 3.2”

- page 4, the function $z$ in Theorem 3.1 is defined in Lemma 3.4 after its first appearance. I suggest giving the definition first.

- page 7, first column, $/N\delta$ -> $/(N\delta)$

- page 7, first column, line 374, budget $B$ -> the budget $B$

- page 8, first column, line 417-419, “search” -> “searches” “maintain” -> “maintains”

- page 8, second column, line 400, “, additional” -> “. Additional”

- page 8, second column, line 407, “performs best” -> “performs the best” Please check throughout the paper, e.g., the same issue in page 12.

- page 8, second column, line 413, “, shows” -> “, which shows”

- page 9, first column, line 449, “EA POMC” -> “EA, POMC”

- page 11, line 573, “Combing” -> “Combining”

- page 11, line 587, “,and” -> “, and”

- page 12, “conclude those” ?

- caption of Table 10, “the probability of each 0.05” ?

- page 16, line 831, “the optimal objective value” -> “the best objective value”

**Other Strengths And Weaknesses:**

Overall, this is a solid work with both theoretical analysis and empirical evaluation. It makes a good contribution to solving the significant problem of subset selection.

**Questions For Authors:**

1. In the introduction part, you mentioned “While there are greedy algorithms that obtain the theoretical optimal approximation of 1 − 1/e, they are generally impractical due to high computational costs” I’d like to see more explanations. Why are these algorithms impractical?

2. The proposed EPOL algorithm enumerates all the $n$ residual problems by excluding each single item. But in the experiments, only a subset of residual problems corresponding to the items with large $f$ values are enumerated. The authors did not give any explanation. I can understand this setting is sufficient to show the superiority of the proposed EPOL as the implemented version is weaker. But I still would like to see the performance of the full version of EPOL. Can it bring further improvement?

3. Why is the objective evaluation noisy for the application of influence maximization?

**Relation To Broader Scientific Literature:**

The studied problem, subset selection, has wide applications across various areas such as machine learning, data mining, and computer networks. This paper proposes a new evolutionary algorithm with better theoretical guarantees and empirical performance. Another contribution of this work is improving the approximation bound of the existing evolutionary algorithm (which performed the best empirically for the subset selection problem). As far as I can tell, this work will bring some influence to the topic of both subset selection and evolutionary computation.

**Theoretical Claims:**

I have checked all the proofs (including the proof of a lemma in the appendix) carefully. The improved approximation guarantee of POMC mainly lies in a tighter analysis of the objective improvement by adding a single item. The proof is mainly accomplished by iteratively adding proper single items. The approximation bound of EPOL is proved by considering the residual problem corresponding to the item having the largest objective value in the optimal solution, which must be generated by enumerating all residual problems.

I almost did not find any mistakes. The proofs are presented clearly. I have only some minor questions or suggestions.

- page 5, first column, line 248, I suggest using $J \geq i+c(v^*)$, to be consistent with the presentation in cases (2) and (3).
- page 5, second column, “This implies that $J_{\max}$ has already exceeded $i$” I suggest adding more explanations. Why the formulas in cases (1)-(3) satisfy Equation (6) with $i+c(v^*)$?

- page 5, second column, “After including … $J_{\max} \geq i+c(v^*)$” Did you consider $i+c(v^*) > B-c(o_c)$, which will contradict with $ J_{\max} \leq B-c(o_c)$?

---

> ### Author Rebuttal · Authors · 2025-04-01
>
> Thank you for your thorough review and valuable suggestions. We sincerely appreciate your feedback, which will help us improve our manuscript. Below are our detailed responses to your queries.
>
> ---
>
> > ... more explanations. Why the formulas in cases (1)-(3) satisfy Equation (6) with $i +c(v^*)$?
>
> Thank you for your suggestion. We will include a more detailed explanation after analyzing cases (1)–(3) as follows:
>
> By combining the inequalities derived from cases (1)–(3), we can conclude that the new solution $X' = X \cup v^*$ satisfies:
>
> $f(X') \geq \left(1 - e^{-\frac{\min\\{i+c(v^*), J\\}}{B-c(o_c)}}\right) \cdot \left(f(X^*) - f(o_c)\right) + \frac{\max\\{i+c(v^*)-J, 0\\}}{B-c(o_c)} \cdot z(r) \cdot f(X^*)$.
>
> Additionally, the cost of solution $X'$ satisfies $c(X') = c(X) + c(v^*) \leq i + c(v^*)$. Consequently, the solution $X'$ must be included in $P$; otherwise, $X'$ must be dominated by one solution in $P$ (line 6 of Algorithm 1). This  implies that $J_{max}$  has already exceeded $i$, contradicting the assumption that $J_{max} = i$. Hence, after including $X'$, we have $J_{max} \geq i + c(v^*)$.
>
> ---
>
> > ... Did you consider $i+c(v^*)>B-c(o_c)$, which will contradict with $J_{max}\leq B-c(o_c)$?
>
> Yes, we did consider this case. On page 5, second column, starting from line 251, we addressed the situation where the inclusion of the new solution $X'$ causes $J_{max} + c(v^*)$ to exceed $B - c(o_c)$, i.e., $J_{max} \leq B - c(o_c) < J_{max} + \delta \leq J_{max} + c(v^*)$. In this situation, we can deduce that $X'$ is a $(1 - z(r))$-approximation solution. We will further clarify this point in the revised manuscript. Thank you.
>
> ---
>
> > “... greedy algorithms that obtain the theoretical optimal approximation of 1 − 1/e” ... Why are these algorithms impractical?
>
> [Sviridenko, 2004] developed a $(1-1/e)$-approximation algorithm by selecting three optimal elements and using a greedy approach, but it has a high time complexity of $O(n^5)$. Later, [Badanidiyuru & Vondrák, 2014] and [Ene & Nguyen, 2019] proposed greedy-based algorithms that achieve a $(1-1/e-\epsilon)$-approximation. However, their computational costs remain prohibitively high, with time complexities of $O(n^2(\frac{\log n}{\epsilon})^{O(1/\epsilon^8)})$ and $(1/\epsilon)^{O(1/\epsilon^4)}n\log^2n$, respectively. These complexities render the algorithms impractical for real-world applications. These details will be clarified in the introduction. Thank you.
>
> ---
>
> > ...only a subset of residual problems corresponding to the items with large values are enumerated. ... But I still would like to see the performance of the full version of EPOL. Can it bring further improvement?
>
> In our experiments, we enumerated a subset of residual problems to balance computational efficiency and performance. We agree that evaluating the full version of EPOL, is both interesting and necessary. Thanks to your suggestion, we conducted additional experiments comparing the original EPOL (as in the manuscript) and the full version (EPOL-full), which enumerates all residual problems. For $q = 5$, the objective values (avg $\pm$ std) on maximum coverage for three datasets are summarized in the tables. For each budget $B$, the larger value is bolded, and ‘$\bullet$’ indicates that EPOL-full significantly outperforms EPOL (Wilcoxon signed-rank test, confidence level 0.05). The results highlight EPOL-full’s potential to improve performance by addressing all residual problems, consistently outperforming EPOL with significant advantages in several cases.
>
> We will revise the manuscript to include these results and discussions. Thank you for your valuable feedback.
>
> | Budget $B$| $300$| $350$  | $400$ |   $450$|$500$  |
> | - | :--: | :--: | :--: | :--: | :--: |
> | **frb-30-15-1:** |
> |EPOL|301.1 $\pm$ 1.0 $\bullet$| 329.7 $\pm$ 0.5 $\bullet$ |354.4 $\pm$ 0.9|375.2 $\pm$ 1.5 $\bullet$|394.1 $\pm$ 1.8|
> |EPOL-full|**302.1 $\pm$ 0.8**|**330.9 $\pm$ 0.3**|**354.8 $\pm$ 0.4**|**377.3 $\pm$ 1.3**|**395.2 $\pm$ 1.1**|
> | **frb-35-17-1:** |
> |EPOL|319.1 $\pm$ 0.8|356.6 $\pm$ 0.9 $\bullet$|389.8 $\pm$ 0.6 $\bullet$|419.0 $\pm$ 0.6| 445.7 $\pm$ 0.6 $\bullet$|
> |EPOL-full|**319.8 $\pm$ 0.4**|**357.9 $\pm$ 0.3**|**390.6 $\pm$ 0.5**|**419.4 $\pm$ 0.3**|**446.5 $\pm$ 0.5**|
> | **congress:** |
> |EPOL|332.8 $\pm$ 0.4|358.0 $\pm$ 0.6 $\bullet$|381.2 $\pm$ 0.6|399.9 $\pm$ 0.7|415.5 $\pm$ 0.5 $\bullet$|
> |EPOL-full|**333.4 $\pm$ 0.5**|**359.0 $\pm$ 0.6**|**381.4 $\pm$ 0.7**|**400.2 $\pm$ 1.0**|**416.1 $\pm$ 0.5**|
>
> ---
>
> > Why is the objective evaluation noisy for the application of influence maximization?
>
> The objective evaluation for influence maximization, i.e., $E[|IC(X)|]$, is noisy because the propagation process is randomized, and we use the average of multiple Monte Carlo simulations to estimate the expectation. Specifically, starting from a solution $X$, we simulate the propagation 500 times and use the average as the estimated objective value.  We will revise to make it clear. Thank you.

---

> > ### Comment · Reviewer_JmGW · 2025-04-02
> >
> > Thanks for the detailed response!
> >
> > My concerns have been addressed well. I will keep my score, and recommend accepting this paper.

---

### Official Review · Reviewer_b7Tq · 2025-03-14

**Overall Recommendation:** 2

**Summary:**

This paper addresses monotone submodular maximization under a linear cost constraint using evolutionary algorithms. It reanalyzes the Pareto Optimization Algorithm for Monotone Cost functions (POMC), providing an improved approximation ratio of $1/2$. Additionally, the authors propose a novel multi-objective evolutionary algorithm, EPOL, which achieves the best-known practical approximation ratio of $0.6174$. Empirical evaluations on maximum coverage and influence maximization tasks demonstrate the superior performance of EPOL compared to existing methods.

**Claims And Evidence:**

The claims made in the submission are supported by clear and convincing evidence

**Essential References Not Discussed:**

No

**Experimental Designs Or Analyses:**

Empirical evaluation is conducted on maximum coverage and influence maximization with six datasets. The results are sound.

**Methods And Evaluation Criteria:**

Proposed methods and evaluation criteria make sense for the problem or application at hand.

**Other Comments Or Suggestions:**

- A linear cost constraint is typically referred to as a knapsack constraint in the literature.

- The paper does not provide a clear definition of what a multi-objective EA is in the introduction or preliminary.

- The core idea behind improving the approximation ratio of POMC should be introduced at a high level at the beginning of Section 3 to provide readers with a clear understanding of the approach before delving into technical details.

**Other Strengths And Weaknesses:**

In this work, the authors re-analyze POMC and demonstrate that it achieves a $1/2$ approximation ratio. By repeatedly running POMC with a 1-guess technique, this approximation ratio is further improved to $0.6174$. However, the technical contribution of the paper appears to be somewhat limited, as the primary advancements rely on modifications and repeated applications of existing methods rather than introducing fundamentally new algorithmic ideas.

Moreover, the algorithm sections consist primarily of lengthy proofs without sufficient discussion of the high-level ideas. This makes it challenging for readers to grasp the intuition behind the proposed methods. The presentation of the paper could be significantly improved by including more intuitive explanations and balancing technical details with conceptual insights.

**Questions For Authors:**

No

**Relation To Broader Scientific Literature:**

The paper studies a fundamental combinatorial optimization problem: monotone submodular maximization under knapsack constraints. It improves the state-of-the-art approximation ratio achieved by evolutionary algorithms (EAs) from $(1-1/e)/2$ to $0.6174$, marking a significant advancement in the field.

**Theoretical Claims:**

No apparent issue found.

---

> ### Author Rebuttal · Authors · 2025-04-01
>
> Thank you for reviewing our paper. We greatly appreciate your time and thoughtful feedback. Below are our detailed responses.
>
> ---
>
> > The core idea behind improving the approximation ratio of POMC should be introduced at a high level at the beginning of Section 3 ...
>
> > Moreover, the algorithm sections consist primarily of lengthy proofs without sufficient discussion of the high-level ideas...
>
> Thank you for your suggestion. In the original paper, we included a high-level explanation of the core idea behind improving POMC's approximation ratio following the proof of Theorem 3.1 (Page 5, left column, lines 307–316). Previous analysis of POMC used a coarse-grained manner to evaluate the lower bound for improving $f(X)$. This led to POMC being able to derive a solution that satisfies $f(X_1) \geq (1 - 1/e)\cdot f(X^*)$, which is, however, infeasible, i.e., $c(X_1) > B$. A connection was then established between $X_1$ and two feasible solutions $X$ and $Y$, demonstrating that $\max\\{f(X), f(Y)\\} \geq f(X_1)/2 \geq (1/2)(1 - 1/e)\cdot f(X^*)$. This weakens the tightness of the bound. In contrast, our analysis adopts a fine-grained approach to evaluate the lower bound for improving $f(X)$ (the analysis in Lemma 3.3 and the careful design of $J_{max}$ in Lemma 3.2), showing that POMC can find a feasible solution $X_2$ with $f(X_2) \geq (1/2) \cdot f(X^*)$ and $c(X_2) \in (B - c(o_c), B]$. Thanks to your suggestion, we will move this discussion to the beginning of Section 3, which will provide readers with a clearer conceptual overview before delving into technical details. Thank you very much.
>
> To ensure clarity and readability, we have included brief explanations of each theorem and lemma, along with proof sketches:
>
> - Page 4, left column, lines 198–203: Connection between Theorem 3.1 and Lemma 3.2; Lemma 3.2's role in proving Theorem 4.1.
> - Page 4, left column, lines 211–219: Relationships between Lemma 3.2 and Lemmas 3.3–3.4; their importance in proofs.
> - Page 4, right column, lines 182–189: Core proof idea of Lemma 3.2: Relies on Lemma 3.3 to analyze $J_{max}$ growth and expected iterations.
>
> - Page 5, left column, lines 279–283: Proof sketch for Theorem 3.1.
> - Page 5, left column, lines 307–316: Main idea for improving POMC’s approximation ratio.
> - Page 5, right column, lines 293–299: Proof sketch for Theorem 4.1.
>
> We believe these explanations provide readers with the necessary intuition to better understand the technical details. Thanks to your suggestion, we will revise to add more intuitive explanations to improve the readability of our work.
>
> ---
>
> > However, the technical contribution of the paper appears to be somewhat limited, as the primary advancements rely on modifications and repeated applications of existing methods rather than introducing fundamentally new algorithmic ideas.
>
> Thank you for your feedback. We would like to clarify that the proposed EPOL algorithm is not a simple repeated application of existing methods. EPOL decomposes the original problem into multiple residual problems. For each $v\in V$, it creates a residual problem $(V \setminus v, f(\cdot \mid v), c, B - c(v))$, which is then solved in parallel using POMC. We also compared EPOL with P-POMC (i.e., a simple repetition of POMC, as detailed in Appendix B.4), where POMC directly executes the original problem in parallel, and the best solution is selected as the final output. The experimental results show that EPOL outperforms P-POMC, thereby highlighting the importance of our design over simple repetition.
>
> In fact, EPOL is carefully designed to analyze the residual problem anchored at the key element $o_f$, defined as $(V \setminus o_f, f(\cdot \mid o_f), c, B - c(o_f))$. It leverages POMC's 1/2-approximation guarantee on the residual problem and connects this guarantee to the original problem, thereby further improving the theoretical bound. Consequently, EPOL not only advances theoretical insights but also attains a SOTA practical performance guarantee.
>
> From a practical problem-solving perspective, the primary goal is to design an algorithm that brings measurable performance improvements. We understand and respect the high expectations for fundamentally new algorithmic ideas. While such breakthroughs are exciting, most research focuses on advancing existing methods to achieve measurable improvements. To our understanding, exploring the potential of existing algorithms and further developing them into approaches that yield improvements in both theoretical analysis and empirical performance is, in itself, a significant contribution.
>
> We hope these clarifications address your concerns. Additionally, we will revise the paper to improve presentation, including defining multi-objective EAs in the introduction, adding intuitive proof explanations, and clarifying that a linear cost constraint refers to a knapsack constraint.
>
> Please do not hesitate to reach out if you have further questions or suggestions. Thank you very much.

---

### Decision · Program_Chairs · 2025-05-01

**Decision:**

Accept (poster)

**Comment:**

This work studies evolutionary algorithms (EAs) for monotone submodular maximization subject to a linear cost constraint. In particular, it:
- improves the analysis of the approximation ratio for the POMC EA [Qian et al., IJCAI 2017] from $1/2 \cdot (1 - 1/e)$ to $0.5$,
- proposes a new multi-objective EA called EPOL, which achieves an approximation ratio of $0.6174$, matching that of the best-known (combinatorial) greedy algorithm.
The paper is well written, gives clean proofs for all results, and presents a compelling set of experiments.

The majority of reviewers are in favor of "accept" (scores: [2, 4, 4, 4]). The reviewer who recommended "weak reject" did not respond to the authors' rebuttal, so our final recommendation for this paper is acceptance.